# Principled Algorithms for Optimizing Generalized Metrics in Binary Classification

Anqi Mao[1]  Mehryar Mohri[2 1]  Yutao Zhong[2]

## Abstract

In applications with significant class imbalance or asymmetric costs, metrics such as the $F_\beta$-measure, AM measure, Jaccard similarity coefficient, and weighted accuracy offer more suitable evaluation criteria than standard binary classification loss. However, optimizing these metrics present significant computational and statistical challenges. Existing approaches often rely on the characterization of the Bayes-optimal classifier, and use threshold-based methods that first estimate class probabilities and then seek an optimal threshold. This leads to algorithms that are not tailored to restricted hypothesis sets and lack finite-sample performance guarantees. In this work, we introduce principled algorithms for optimizing generalized metrics, supported by $\mathcal{H}$-consistency and finite-sample generalization bounds. Our approach reformulates metric optimization as a generalized cost-sensitive learning problem, enabling the design of novel surrogate loss functions with provable $\mathcal{H}$-consistency guarantees. Leveraging this framework, we develop new algorithms, METRO (*Metric Optimization*), with strong theoretical performance guarantees. We report the results of experiments demonstrating the effectiveness of our methods compared to prior baselines.

## 1. Introduction

In many applications, performance metrics such as the $F_\beta$-*measure* (Lewis, 1995a; Jansche, 2005; Ye et al., 2012), the *AM measure* (Menon et al., 2013), the *Jaccard similarity*

[1]Courant Institute of Mathematical Sciences, New York, NY; [2]Google Research, New York, NY. Correspondence to: Anqi Mao <aqmao@cims.nyu.edu>, Mehryar Mohri <mohri@google.com>, Yutao Zhong <yutaozhong@google.com>.

*Proceedings of the 42$^{nd}$ International Conference on Machine Learning*, Vancouver, Canada. PMLR 267, 2025. Copyright 2025 by the author(s).

*coefficient* (Sokolova & Lapalme, 2009), or the *weighted accuracy* are preferred over the standard zero-one misclassification loss. These metrics are particularly relevant in scenarios with significant class imbalance, such as fraud detection, medical diagnosis, and information retrieval (Lewis, 1995b; Drummond & Holte, 2005; He & Garcia, 2009; Gu et al., 2009), or in settings where classification costs are asymmetrical. However, optimizing these alternative metrics presents both computational and statistical challenges, as they deviate from the standard single-loss expectation framework commonly used in surrogate loss function analysis (Steinwart, 2007).

Research on the design of algorithms for specific instances of such metrics or the general family has largely focused on the characterization of the Bayes-optimal classifier. For the standard zero-one binary classification loss, it is known that the Bayes classifier can be defined as $x \mapsto \text{sign}(\eta(x) - \frac{1}{2})$, where $\eta(x)$ is the probability of a positive label conditioned on $x$. A similar characterization applies to the weighted zero-one loss (Scott, 2012), where the threshold corresponds to a weight different from $\frac{1}{2}$. For imbalanced metrics, Ye et al. (2012) characterized the Bayes-classifier for the $F_1$-measure, while Menon et al. (2013) characterized the Bayes classifier for the AM measure as $x \mapsto \text{sign}(\eta(x) - \delta^*)$ for some $\delta^* \in (0, 1)$. These results were later generalized by Koyejo et al. (2014) to a broader family of metrics that can be formulated as a ratio of two linear functions of true positive (TP), false positive (FP), true negative (TN), and false negative (FN) statistics. This family of linear-fractional metrics includes the aforementioned weighted accuracy, $F_\beta$-measures, the AM measure, as well as the *Jaccard similarity coefficient* (Sokolova & Lapalme, 2009), among others.

Existing algorithms for optimizing these metrics heavily rely on the structure of the Bayes-optimal solution (Koyejo et al., 2014; Parambath et al., 2014). These methods generally adopt a two-stage approach: first estimating the conditional probability $\eta(x)$ and then searching for a suitable threshold $\delta^*$ specific to the metric.

These algorithms naturally come with consistency guarantees (Koyejo et al., 2014). However, consistency is an asymptotic property and does not provide explicit convergence rate guarantees. Moreover, it applies only to the

overly broad class of all measurable functions, making it less relevant in practical scenarios where learning is constrained to a restricted hypothesis class (Long & Servedio, 2013; Zhang & Agarwal, 2020).

Beyond these theoretical limitations, a key drawback of prior work is its reliance on the structure of the Bayes-optimal classifier, which may differ significantly from the best predictor within a given hypothesis class. For example, we show that in a two-dimensional setting with linear classifiers (see Section 5, Figure 1), the best linear classifier derived via a margin-maximizing classifier can have a significantly different orientation from the best linear classifier optimized for an $F_\beta$-measure.

**Contributions.** In contrast to previous work, we introduce a hypothesis set-specific analysis grounded in recent advances in $\mathcal{H}$-*consistency bounds* (Awasthi, Mao, Mohri, and Zhong, 2022a;b; Mao, Mohri, and Zhong, 2023f; 2024i). Unlike Bayes-consistency, these bounds are non-asymptotic and explicitly account for the hypothesis set $\mathcal{H}$ used in practice. They provide direct upper bounds on the target estimation error in terms of the surrogate estimation error, making them more relevant to practical learning. We leverage this framework to develop new algorithms with strong theoretical guarantees, including finite-sample learning bounds.

**Related work**. A comprehensive discussion on consistency and generalized metrics in binary classification is provided in Appendix A. Here, we briefly summarize the most relevant prior work. Koyejo et al. (2014) studied a broad family of performance metrics, including the $F_\beta$-measure, AM measure, Jaccard similarity coefficient, and weighted accuracy. They proposed two-stage thresholding algorithms that first train a binary classifier (e.g., logistic regression) to estimate $\eta(x)$ and then optimize a threshold $\theta$ to maximize the empirical metric of interest. Their approach comes with a consistency-type guarantee but lacks finite-sample guarantees. Parambath et al. (2014) focused specifically on optimizing the $F_1$-measure. Their algorithm is similar to the second approach in Koyejo et al. (2014) but additionally considers an extra threshold $\theta'$, both of which are optimized to maximize the empirical metric. They provide stability-type guarantees, though their analysis lacks explicit details.

**Structure of the paper.** We introduce principled algorithms for optimizing a broad family of generalized metrics, supported by $\mathcal{H}$-consistency bounds and finite-sample generalization bounds. We first present an equivalent reformulation of the problem of minimizing generalized metrics (Section 3). This reformulation allows us to interpret the problem as minimizing a generalized cost-sensitive target loss function (Section 4.1). We then address this broader cost-sensitive learning problem by introducing a new family of surrogate loss functions tailored to this framework

(Section 4.2). We further establish strong $\mathcal{H}$-consistency guarantees for these surrogate losses (Section 4.3). In Section 5, we leverage these theoretical insights to design new algorithms for optimizing generalized metrics, METRO (*Metric Optimization*), for which we prove strong performance guarantees. Finally, we present experimental results in Section 6, demonstrating the effectiveness of our algorithms in comparison to prior baselines.

## 2. Preliminaries

**Binary classification**. We consider the familiar setting of binary classification, where the input space is denoted by $\mathcal{X}$, the label space by $\mathcal{Y} = \{+1, -1\}$, and the data is distributed according to an unknown distribution $\mathcal{D}$ over $\mathcal{X} \times \mathcal{Y}$. We will consider prediction functions $h$ mapping from $\mathcal{X}$ to $\mathbb{R}$ and will denote by $\mathcal{H}_{\text{all}}$ the family of all measurable functions of this type. For a given loss function $\ell$ mapping from $\mathcal{H}_{\text{all}} \times \mathcal{X} \times \mathcal{Y}$ to $\mathbb{R}$, the *expected loss* of a hypothesis $h$ and the *best-in-class expected loss* of a hypothesis set $\mathcal{H} \subseteq \mathcal{H}_{\text{all}}$ are defined by $\mathcal{E}_\ell(h) = \mathbb{E}_{(x,y)\sim\mathcal{D}}[\ell(h, x, y)]$ and $\mathcal{E}_\ell^*(\mathcal{H}) = \inf_{h \in \mathcal{H}} \mathcal{E}_\ell(h)$. The *excess error* of a hypothesis $h$, $\mathcal{E}_\ell(h) - \mathcal{E}_\ell^*(\mathcal{H}_{\text{all}})$, can be decomposed as the sum of its *estimation error*, $\mathcal{E}_\ell(h) - \mathcal{E}_\ell^*(\mathcal{H})$, and the approximation error of $\mathcal{H}$, $\mathcal{E}_\ell^*(\mathcal{H}) - \mathcal{E}_\ell^*(\mathcal{H}_{\text{all}})$. Given a sample $S = (x_1, \ldots, x_m)$ and a hypothesis $h$, the *empirical error* is defined by $\widehat{\mathcal{E}}_{\ell,S}(h) = \frac{1}{m}\sum_{i=1}^m \ell(h, x_i, y_i)$.

**Consistency guarantees**. Given a surrogate loss function $\ell_1$ and a target loss function $\ell_2$, a fundamental property of $\ell_1$ with respect to $\ell_2$ is *Bayes-consistency* (Zhang, 2004; Bartlett et al., 2006; Steinwart, 2007).

**Definition 2.1** (**Bayes-Consistency**). A loss function $\ell_1$ is *Bayes-consistent* with respect to a loss function $\ell_2$ if, for all distributions and sequences $\{h_n\}_{n\in\mathbb{N}} \subset \mathcal{H}_{\text{all}}$, if $[\mathcal{E}_{\ell_1}(h_n) - \mathcal{E}_{\ell_1}^*(\mathcal{H}_{\text{all}})]$ tends to zero when $n$ tends to $+\infty$, then $[\mathcal{E}_{\ell_2}(h_n) - \mathcal{E}_{\ell_2}^*(\mathcal{H}_{\text{all}})]$ also tends to zero.

While Bayes consistency is a natural and desirable property, it is inherently asymptotic and applies only to the family of all measurable functions. As such, it offers no insights into convergence rates or the behavior of learning algorithms when restricted hypothesis classes are used, as is typical in machine learning applications. In this context, a more informative property is an $\mathcal{H}$-*consistency bound* (Awasthi, Mao, Mohri, and Zhong, 2022a; Mao, Mohri, and Zhong, 2023f; Zhong, 2025), which provides a quantitative guarantee tailored to specific hypothesis classes.

**Definition 2.2** ($\mathcal{H}$-**Consistency bound**). A surrogate loss $\ell_1$ admits an $\mathcal{H}$-*consistency bound* with respect to the target loss $\ell_2$ if there exists a non-decreasing concave function $\Gamma: \mathbb{R}_+ \to \mathbb{R}_+$ with $\Gamma(0) = 0$ such that, for all $h \in \mathcal{H}$ and all

distributions, the following inequality holds:

$$\mathcal{E}_{\ell_2}(h) - \mathcal{E}_{\ell_2}^*(\mathcal{H}) + \mathcal{M}_{\ell_2}(\mathcal{H})$$
$$\leq \Gamma\big(\mathcal{E}_{\ell_1}(h) - \mathcal{E}_{\ell_1}^*(\mathcal{H}) + \mathcal{M}_{\ell_1}(\mathcal{H})\big), \quad (1)$$

where, for any loss function $\ell$, $\mathcal{M}_\ell(\mathcal{H})$ is defined as $\mathcal{M}_\ell(\mathcal{H}) = \mathcal{E}_\ell^*(\mathcal{H}) - \mathbb{E}[\inf_{h \in \mathcal{H}} \mathbb{E}[\ell(h, x, y) \mid x]]$, and is referred to as the *minimizability gap*.

For $\mathcal{H} = \mathcal{H}_{\text{all}}$, the second term coincides with $\mathcal{E}_\ell^*(\mathcal{H})$ (Steinwart, 2007; Mao, Mohri, and Zhong, 2024i) and the minimizability gap is zero. In this case, the bound simplifies to $\mathcal{E}_{\ell_2}(h) - \mathcal{E}_{\ell_2}^*(\mathcal{H}) \leq \Gamma\big(\mathcal{E}_{\ell_1}(h) - \mathcal{E}_{\ell_1}^*(\mathcal{H})\big)$ which, in particular, implies Bayes-consistency. More generally, the minimizability gap vanishes when $\mathcal{E}_\ell^*(\mathcal{H}) = \mathcal{E}_\ell^*(\mathcal{H}_{\text{all}})$. The minimizability gap is always upper bounded by the approximation error but it can be strictly smaller in many cases. In general, $\mathcal{H}$-consistency bounds provide a stronger, non-asymptotic, and hypothesis set-dependent consistency guarantee.

# 3. Problem Formulation

Our goal is to devise a principled learning algorithm seeking to optimize any instance within a broad family of generalized metrics. We first define this family and then reformulate the learning problem using an alternative loss function.

## 3.1. Generalized Metrics

We study *generalized metrics* for binary classification, defined as follows for any $h \in \mathcal{H}_{\text{all}}$:

$$\mathcal{L}(h) = \frac{\mathbb{E}_{(x,y) \sim \mathcal{D}}[\alpha_1 \mathsf{h}(x)y + \alpha_2 y + \alpha_3 \mathsf{h}(x) + \alpha_4]}{\mathbb{E}_{(x,y) \sim \mathcal{D}}[\beta_1 \mathsf{h}(x)y + \beta_2 y + \beta_3 \mathsf{h}(x) + \beta_4]}, \quad (2)$$

where $\mathsf{h}(x) = \text{sign}(h(x))$ represents the prediction of a hypothesis $h$ on an input $x \in \mathcal{X}$, with $\text{sign}(\alpha) = 1_{\alpha \geq 0} - 1_{\alpha < 0}$, and with $\boldsymbol{\alpha} = [\alpha_1, \alpha_2, \alpha_3, \alpha_4], \boldsymbol{\beta} = [\beta_1, \beta_2, \beta_3, \beta_4] \in \mathbb{R}^4$. To avoid notational clutter, we assume the dependence on $(\boldsymbol{\alpha}, \boldsymbol{\beta})$ is understood from the context and omit it when it is not necessary. Note that the true positive (TP), false positive (FP), true negative (TN), and false negative (FN) statistics can be expressed in terms of $\mathbb{E}_{(x,y)}[\mathsf{h}(x)y]$, $\mathbb{E}_{(x,y)}[y]$, and $\mathbb{E}_{(x,y)}[\mathsf{h}(x)]$:

$$\text{TP} = \mathbb{E}_{(x,y)}\big[\overline{\mathsf{h}}(x)\overline{y}\big] \qquad \text{FN} = \mathbb{E}_{(x,y)}\big[1 - \overline{\mathsf{h}}(x) - \overline{y} + \overline{\mathsf{h}}(x)\overline{y}\big]$$

$$\text{TN} = \mathbb{E}_{(x,y)}\big[\overline{y} - \overline{\mathsf{h}}(x)\overline{y}\big] \quad \text{FP} = \mathbb{E}_{(x,y)}\big[\overline{\mathsf{h}}(x) - \overline{\mathsf{h}}(x)\overline{y}\big],$$

where $\overline{\mathsf{h}}(x) = \frac{\mathsf{h}(x)+1}{2} \in \{0,1\}$ and $\overline{y} = \frac{y+1}{2} \in \{0,1\}$. Thus, any ratio of two linear combinations of TP, FP, TN, and FN, which are considered in (Koyejo et al., 2014), can be expressed equivalently in the form of (2). This formulation

covers several widely used metrics, including the AM measure (Menon et al., 2013), the $F_\beta$-measure (Ye et al., 2012), the Jaccard similarity coefficient (JAC) (Sokolova & Lapalme, 2009), Weighted Accuracy (WA), and many others of interest. Given a hypothesis set $\mathcal{H}$, our goal is to find a hypothesis $h \in \mathcal{H}$ with small loss $\mathcal{L}(h)$. We denote the best-in-class expected loss by $\mathcal{L}^*(\mathcal{H}) = \inf_{h \in \mathcal{H}} \mathcal{L}(h)$. Given a sample $S = (x_1, \ldots, x_m)$ and a hypothesis $h$, the *empirical loss* is defined by $\widehat{\mathcal{L}}_S(h) = \frac{\frac{1}{m}\sum_{i=1}^m[\alpha_1 \mathsf{h}(x_i)y_i + \alpha_2 y_i + \alpha_3 \mathsf{h}(x_i) + \alpha_4]}{\frac{1}{m}\sum_{i=1}^m[\beta_1 \mathsf{h}(x_i)y_i + \beta_2 y_i + \beta_3 \mathsf{h}(x_i) + \beta_4]}$.

## 3.2. Equivalent Problem

The generalized metrics introduced above are defined as the ratio of two expected loss functions. This formulation differs from the more familiar single-loss expectations, which are commonly used for deriving and analyzing surrogate loss functions in machine learning. In this section, we present an equivalent reformulation of the problem of minimizing $\mathcal{L}(h)$. This reformulation will provide a more convenient framework for designing surrogate loss functions.

For any $(h, x, y) \in \mathcal{H}_{\text{all}} \times \mathcal{X} \times \mathcal{Y}$, we define two loss functions: $\ell_{\boldsymbol{\alpha}} : (h, x, y) \mapsto \alpha_1 \mathsf{h}(x)y + \alpha_2 y + \alpha_3 \mathsf{h}(x) + \alpha_4$ and $\ell_{\boldsymbol{\beta}} : (h, x, y) \mapsto \beta_1 \mathsf{h}(x)y + \beta_2 y + \beta_3 \mathsf{h}(x) + \beta_4$. Using these definitions, $\mathcal{L}(h)$ can be rewritten as follows:

$$\mathcal{L}(h) = \frac{\mathbb{E}_{(x,y) \sim \mathcal{D}}[\ell_{\boldsymbol{\alpha}}(h, x, y)]}{\mathbb{E}_{(x,y) \sim \mathcal{D}}[\ell_{\boldsymbol{\beta}}(h, x, y)]},$$

that is, the fractional form of the expected losses of $\ell_{\boldsymbol{\alpha}}$ and $\ell_{\boldsymbol{\beta}}$. Next, to minimize $\mathcal{L}(h)$, we reformulate it as an equivalent optimization problem, which will facilitate further analysis and surrogate design.

For any $\lambda$ (and $\boldsymbol{\alpha}$ and $\boldsymbol{\beta}$), define the loss function $\ell^\lambda$ by:

$$\forall (h, x, y), \ell^\lambda(h, x, y) = \ell_{\boldsymbol{\alpha}}(h, x, y) - \lambda \ell_{\boldsymbol{\beta}}(h, x, y). \quad (3)$$

We will denote by $\widehat{\mathcal{E}}_{\ell^\lambda, S}$ the empirical loss of $\ell^\lambda$ over a sample $S$. Without loss of generality, we assume throughout that $\mathbb{E}_{(x,y) \sim \mathcal{D}}[\ell_{\boldsymbol{\beta}}(h, x, y)]$ is positive for all $h \in \mathcal{H}$. If this condition does not hold, it can be enforced by redefining $\ell_{\boldsymbol{\beta}} = -\ell_{\boldsymbol{\beta}}$ and $\ell_{\boldsymbol{\alpha}} = -\ell_{\boldsymbol{\alpha}}$, as needed. For convenience, we will further define $\underline{\ell}_\beta = \inf_{h \in \mathcal{H}} \mathbb{E}_{(x,y) \sim \mathcal{D}}[\ell_{\boldsymbol{\beta}}(h, x, y)]$ and assume $\underline{\ell}_\beta > 0$. Similarly, let $\overline{\ell}_\beta = \sup_{h \in \mathcal{H}} \mathbb{E}_{(x,y) \sim \mathcal{D}}[\ell_{\boldsymbol{\beta}}(h, x, y)]$, where $\overline{\ell}_\beta < +\infty$. The following theorem establishes that minimizing $\mathcal{L}(h)$ over $\mathcal{H}$ is equivalent to minimizing $\mathcal{E}_{\ell^{\lambda^*}}(h)$ over $\mathcal{H}$, where $\lambda^* = \mathcal{L}^*(\mathcal{H})$.

**Theorem 3.1.** *The equality $\mathcal{L}(h^*) = \mathcal{L}^*(\mathcal{H})$ holds for $h^* \in \mathcal{H}$ if and only if $\mathcal{E}_{\ell^{\lambda^*}}(h^*) = \mathcal{E}_{\ell^{\lambda^*}}(\mathcal{H}) = 0$.*

*Proof.* Assume that there exists a hypothesis $h^* \in \mathcal{H}$ such that $\mathcal{L}(h^*) = \mathcal{L}^*(\mathcal{H})$ holds. We have for all $h \in \mathcal{H}$:

$$\lambda^* = \mathcal{L}(h^*) = \frac{\mathbb{E}[\ell_{\boldsymbol{\alpha}}(h^*, x, y)]}{\mathbb{E}[\ell_{\boldsymbol{\beta}}(h^*, x, y)]} \leq \frac{\mathbb{E}[\ell_{\boldsymbol{\alpha}}(h, x, y)]}{\mathbb{E}[\ell_{\boldsymbol{\beta}}(h, x, y)]}.$$

Thus, under the assumption, the following holds for all $h \in \mathcal{H}$:

$$\lambda^* \mathop{\mathbb{E}}_{(x,y) \sim \mathcal{D}} [\ell_{\boldsymbol{\beta}}(h, x, y)] \leq \mathop{\mathbb{E}}_{(x,y) \sim \mathcal{D}} [\ell_{\boldsymbol{\alpha}}(h, x, y)],$$

$$\lambda^* \mathop{\mathbb{E}}_{(x,y) \sim \mathcal{D}} [\ell_{\boldsymbol{\beta}}(h^*, x, y)] = \mathop{\mathbb{E}}_{(x,y) \sim \mathcal{D}} [\ell_{\boldsymbol{\alpha}}(h^*, x, y)].$$

This implies that $\mathcal{E}_{\ell^{\lambda^*}}(h^*) = \mathcal{E}_{\ell^{\lambda^*}}(\mathcal{H}) = 0$, which completes one direction of the proof.

Assume now that there exists $h^* \in \mathcal{H}$ such that $\mathcal{E}_{\ell^{\lambda^*}}(h^*) = \mathcal{E}_{\ell^{\lambda^*}}(\mathcal{H}) = 0$ holds. We have for all $h \in \mathcal{H}$:

$$\mathop{\mathbb{E}}_{(x,y) \sim \mathcal{D}} [\ell_{\boldsymbol{\alpha}}(h^*, x, y)] - \lambda^* \mathop{\mathbb{E}}_{(x,y) \sim \mathcal{D}} [\ell_{\boldsymbol{\beta}}(h^*, x, y)] = 0$$

$$\mathop{\mathbb{E}}_{(x,y) \sim \mathcal{D}} [\ell_{\boldsymbol{\alpha}}(h, x, y)] - \lambda^* \mathop{\mathbb{E}}_{(x,y) \sim \mathcal{D}} [\ell_{\boldsymbol{\beta}}(h, x, y)] \geq 0.$$

Thus, we have $\lambda^* = \frac{\mathbb{E}_{(x,y) \sim \mathcal{D}}[\ell_{\boldsymbol{\alpha}}(h^*, x, y)]}{\mathbb{E}_{(x,y) \sim \mathcal{D}}[\ell_{\boldsymbol{\beta}}(h^*, x, y)]} = \mathcal{L}(h^*) \leq \mathcal{L}(h)$ for all $h \in \mathcal{H}$, which implies $\mathcal{L}(h^*) = \mathcal{L}^*(\mathcal{H})$. This completes the proof. □

More generally, the following non-asymptotic equivalence holds. The proof is given in Appendix B.

**Theorem 3.2.** *Fix $\eta \geq 0$ and $h \in \mathcal{H}$. Then, the inequality $\mathcal{E}_{\ell^{\lambda^*}}(h) \leq \eta$ holds if and only if $\mathcal{L}(h) - \mathcal{L}^*(\mathcal{H}) \leq \frac{\eta}{\mathbb{E}_{(x,y) \sim \mathcal{D}}[\ell_{\boldsymbol{\beta}}(h,x,y)]}$.*

Theorems 3.1 and 3.2 show that our problem can be reduced to minimizing the expected value of the loss function $\ell^{\lambda^*}$. However, this remains intractable because $\ell^{\lambda^*}$ is non-differentiable and even non-continuous as a function of $h$ (as it is linear in h). In the next section, we will define a general family of consistent surrogate losses for $\ell^{\lambda^*}$ that are more suitable for practical optimization techniques.

# 4. General Cost-Sensitive Learning

In this section, we first establish that $\ell^{\lambda^*}$ can be interpreted as a generalized cost-sensitive target loss function. We then consider the broader problem of cost-sensitive learning and introduce a novel family of surrogate loss functions tailored to this framework. Finally, we present strong theoretical guarantees for these surrogate losses, demonstrating that minimizing them yields efficient algorithms for both general cost-sensitive learning and the optimization of $\ell^{\lambda^*}$.

## 4.1. General Target Loss

Define $\boldsymbol{\gamma} = [\gamma_1, \gamma_2, \gamma_3, \gamma_4]$, where $\gamma_i = \alpha_i - \lambda^* \beta_i$, for $i \in \{1, 2, 3, 4\}$. Then, for any $h \in \mathcal{H}$ and $(x, y) \in \mathcal{X} \times \mathcal{Y}$, $\ell^{\lambda^*}(h, x, y)$ can be expressed as:

$$\gamma_1 \mathsf{h}(x)y + \gamma_2 y + \gamma_3 \mathsf{h}(x) + \gamma_4 = \mathsf{L}_{\boldsymbol{\gamma}}(\mathsf{h}(x), y), \quad (4)$$

where $\mathsf{L}_{\boldsymbol{\gamma}}$ is a cost-sensitive function over $\mathcal{Y} \times \mathcal{Y}$ defined as

$$\mathsf{L}_{\boldsymbol{\gamma}}(y', y) = \begin{cases} \gamma_1 + \gamma_2 + \gamma_3 + \gamma_4 & y' = 1, y = 1, \\ -\gamma_1 - \gamma_2 + \gamma_3 + \gamma_4 & y' = 1, y = -1, \\ -\gamma_1 + \gamma_2 - \gamma_3 + \gamma_4 & y' = -1, y = 1, \\ \gamma_1 - \gamma_2 - \gamma_3 + \gamma_4 & y' = -1, y = -1. \end{cases}$$

Adding a constant to $\mathsf{L}_{\boldsymbol{\gamma}}$ does not affect its minimization. Therefore, we can augment it with a non-negative constant $\tau$ to ensure $\mathsf{L}_{\boldsymbol{\gamma}} + \tau \geq 0$. A suitable choice is $\tau = |\gamma_1| + |\gamma_2| + |\gamma_3| + |\gamma_4|$.

This can be regarded as a special case of the *general cost-sensitive learning* problem (Elkan, 2001). In this context, we consider a cost-sensitive loss function $\overline{\mathsf{L}} \colon \{+1, -1\} \times \{+1, -1\} \to \mathbb{R}_+$, which defines four non-negative costs: $\overline{\mathsf{L}}(+1, +1)$, $\overline{\mathsf{L}}(+1, -1)$, $\overline{\mathsf{L}}(-1, +1)$, and $\overline{\mathsf{L}}(-1, -1)$. These costs depend on the prediction $\mathsf{h}(x) \in \{-1, +1\}$ and the true label $y \in \{-1, +1\}$. We denote by $\mathsf{L}$ the target loss function induced by $\overline{\mathsf{L}}$, defined as:

$$\forall (h, x, y), \quad \mathsf{L}(h, x, y) = \overline{\mathsf{L}}(\mathsf{h}(x), y). \quad (5)$$

This approach is referred to as general cost-sensitive learning, as opposed to the specific cost-sensitive learning problem analyzed in (Scott, 2012), where $\overline{\mathsf{L}}(+1, -1) = \theta$, $\overline{\mathsf{L}}(-1, +1) = 1 - \theta$, and $\overline{\mathsf{L}}(+1, +1) = \overline{\mathsf{L}}(-1, -1) = 0$. Furthermore, the existing $\theta$-weighted surrogate losses in (Scott, 2012) are not applicable in this general setting.

## 4.2. General Surrogate Losses

As with many target loss functions in learning problems, such as the zero-one loss in binary classification, directly minimizing the general cost-sensitive loss function $\mathsf{L}$ is intractable for most hypothesis sets due to its non-continuity and non-differentiability. Instead, surrogate losses are typically adopted in practice. These surrogate losses are designed to be consistent, or even $\mathcal{H}$-consistent in the standard classification settings (Zhang, 2004; Bartlett et al., 2006; Awasthi et al., 2022a;b; Mao et al., 2023f). They are frequently formulated as margin-based loss functions (Lin, 2004), which are defined by non-increasing functions $\Phi \colon t \to \mathbb{R}$ that upper bound $t \mapsto 1_{t \leq 0}$. For example, $\Phi$ could be the hinge loss, $t \mapsto \max\{0, 1 - t\}$, or the logistic loss, $t \mapsto \log(1 + e^{-t})$.

Here, we define the following surrogate loss functions for the general cost-sensitive loss function $\mathsf{L}$ by extending the standard margin-based loss function to the general cost-sensitive setting:

$$\mathsf{L}_{\Phi}(h, x, y) = \overline{\mathsf{L}}(+1, y)\Phi(-\mathsf{h}(x)) + \overline{\mathsf{L}}(-1, y)\Phi(\mathsf{h}(x)). \quad (6)$$

Table 1 lists common examples of $\Phi$ along with the corresponding general cost-sensitive surrogate losses. A special

*Table 1.* Common Margin-Based Losses and Their General Cost-Sensitive Surrogate Extensions.

| Name | $\Phi(t)$ | Cost-Sensitive Surrogate Loss $\mathsf{L}_\Phi$ |
|------|-----------|------------------------------------------------|
| Exponential | $\Phi_{\exp}(t) = e^{-t}$ | $\overline{\mathsf{L}}(+1,y)e^{h(x)} + \overline{\mathsf{L}}(-1,y)e^{-h(x)}$ |
| Logistic | $\Phi_{\log}(t) = \log\big(1 + e^{-t}\big)$ | $\overline{\mathsf{L}}(+1,y)\log\big(1 + e^{h(x)}\big) + \overline{\mathsf{L}}(-1,y)\log\big(1 + e^{-h(x)}\big)$ |
| Quadratic | $\Phi_{\mathrm{quad}}(t) = \max\{1-t,0\}^2$ | $\overline{\mathsf{L}}(+1,y)\Phi_{\mathrm{quad}}(-h(x)) + \overline{\mathsf{L}}(-1,y)\Phi_{\mathrm{quad}}(h(x))$ |
| Hinge | $\Phi_{\mathrm{hinge}}(t) = \max\{1-t,0\}$ | $\overline{\mathsf{L}}(+1,y)\Phi_{\mathrm{hinge}}(-h(x)) + \overline{\mathsf{L}}(-1,y)\Phi_{\mathrm{hinge}}(h(x))$ |
| Sigmoid | $\Phi_{\mathrm{sig}}(t) = 1 - \tanh(kt), k > 0$ | $\overline{\mathsf{L}}(+1,y)\Phi_{\mathrm{sig}}(-h(x)) + \overline{\mathsf{L}}(-1,y)\Phi_{\mathrm{sig}}(h(x))$ |
| $\rho$-Margin | $\Phi_\rho(t) = \min\big\{1, \max\big\{0, 1 - \frac{t}{\rho}\big\}\big\}, \rho > 0$ | $\overline{\mathsf{L}}(+1,y)\Phi_\rho(-h(x)) + \overline{\mathsf{L}}(-1,y)\Phi_\rho(h(x))$ |

case of (6) arises when the costs satisfy $\overline{\mathsf{L}}(+1,-1) = \theta$, $\overline{\mathsf{L}}(-1,+1) = 1 - \theta$, and $\overline{\mathsf{L}}(+1,+1) = \overline{\mathsf{L}}(-1,-1) = 0$. This corresponds to the $\theta$-weighted surrogate loss considered in (Scott, 2012; Koyejo et al., 2014). Our formulation generalizes and significantly extends this surrogate loss framework to address the broader context of general cost-sensitive learning.

It is important to highlight that our proposed algorithm, detailed in Section 5, for optimizing generalized metrics differs fundamentally from the second algorithm introduced in (Koyejo et al., 2014) (see also (Parambath et al., 2014)), despite both leveraging a sub-algorithm for cost-sensitive learning. As discussed in Section 1, their approach involves approximating the Bayes-classifier, a threshold function, using the $\theta$-weighted cost-sensitive surrogate loss function. In contrast, our algorithm minimizes a *general* cost-sensitive surrogate loss function that is $\mathcal{H}$-consistent (see Section 4.3) with respect to a *general* cost-sensitive target loss function (4), with label-dependent costs that take into account the best-in-class error of the generalized metric in binary classification, which can be approximated through a binary search-based algorithm (see Section 5).

### 4.3. Theoretical Guarantees

In this section, we establish strong theoretical guarantees for a surrogate loss $\mathsf{L}_\Phi$. Specifically, we derive $\mathcal{H}$-consistency bounds for $\mathsf{L}_\Phi$ with respect to the cost-sensitive loss function $\mathsf{L}$, focusing on commonly used hypothesis sets.

We define a hypothesis set $\mathcal{H}$ as *regular* if, for any $x \in \mathcal{X}$, the set of predictions made by the hypotheses in $\mathcal{H}$ on $x$ covers all possible labels: $\{h(x) : h \in \mathcal{H}\} = \{+1, -1\}$. Commonly used hypothesis sets, such as linear models, neural networks, and the family of all measurable functions, all naturally satisfy this regularity condition.

It was shown by Awasthi, Mao, Mohri, and Zhong (2022a) that common margin-based loss functions, such as the hinge loss, logistic loss, and exponential loss, admit strong $\mathcal{H}$-consistency bounds with respect to the binary zero-one loss function $\ell_{0-1} : (h, x, y) \mapsto 1_{h(x) \neq y}$ when using such regular hypothesis sets. The next result shows that, for such margin-based loss functions $\Phi$, their corresponding

cost-sensitive surrogate losses $\mathsf{L}_\Phi$ (Eq. (6)) also admit $\mathcal{H}$-consistency bounds with respect to the cost-sensitive loss $\mathsf{L}$ (Eq. (5)).

**Theorem 4.1.** *Assume that $\mathsf{L}$ takes values in $[0, \mathsf{L}_{\max}]$. Let $\mathcal{H}$ be a regular hypothesis set and $\Phi$ a margin-based loss function for the binary zero-one loss function $\ell_{0-1}$. Assume that $\Phi$ admits a $\Gamma$-$\mathcal{H}$-consistency bound with respect to $\ell_{0-1}$ for a function $\Gamma : t \mapsto \beta t^\alpha$, with $\alpha \in (0, 1]$ and $\beta > 0$. Then, $\mathsf{L}_\Phi$ admits a $\overline{\Gamma}$-$\mathcal{H}$-consistency bound with respect to $\mathsf{L}$, where $\overline{\Gamma}(t) = \beta(2\mathsf{L}_{\max})^{1-\alpha} t^\alpha$.*

The proof is included in Appendix C. Based on the results of Awasthi et al. (2022a), the theorem holds with $\Gamma(t) = \sqrt{2t}$ for the logistic loss and the exponential loss ($\alpha = 1/2$), $\Gamma(t) = \sqrt{t}$ for the quadratic loss, and $\Gamma(t) = t$ for the hinge loss, sigmoid loss and $\rho$-margin loss ($\alpha = 1$).

As already mentioned, when the best-in-class error coincides with the Bayes error, $\mathcal{E}_\ell^*(\mathcal{H}) = \mathcal{E}_\ell^*(\mathcal{H}_{\mathrm{all}})$ for $\ell = \mathsf{L}_\Phi$ and $\ell = \mathsf{L}$, the minimizability gaps $\mathcal{M}_{\mathsf{L}}(\mathcal{H})$ and $\mathcal{M}_{\mathsf{L}_\Phi}(\mathcal{H})$ vanish. Under these conditions, the $\mathcal{H}$-consistency bound guarantees that when the surrogate estimation error $\mathcal{E}_{\mathsf{L}_\Phi}(h) - \mathcal{E}_{\mathsf{L}_\Phi}^*(\mathcal{H})$ is reduced to $\epsilon$, the estimation error of the cost-sensitive loss $\mathcal{E}_{\mathsf{L}}(h) - \mathcal{E}_{\mathsf{L}}^*(\mathcal{H})$ is upper bounded by $\overline{\Gamma}(\epsilon)$.

More generally, since a concave function $\Gamma$ is sub-additive over $\mathbb{R}_+$, the following guarantee holds:

$$\mathcal{E}_{\mathsf{L}}(h) - \mathcal{E}_{\mathsf{L}}^*(\mathcal{H}))$$
$$\leq \overline{\Gamma}\big(\mathcal{E}_{\mathsf{L}_\Phi}(h) - \mathcal{E}_{\mathsf{L}_\Phi}^*(\mathcal{H})\big) + \overline{\Gamma}(\mathcal{M}_{\mathsf{L}_\Phi}(\mathcal{H})) - \mathcal{M}_{\mathsf{L}}(\mathcal{H}).$$

When the minimizability gaps (or the upper bounding approximation errors) are small, the last terms, $\big[\overline{\Gamma}(\mathcal{M}_{\mathsf{L}_\Phi}(\mathcal{H})) - \mathcal{M}_{\mathsf{L}}(\mathcal{H})\big]$ is also small and close to zero. In particular, when $\mathcal{H} = \mathcal{H}_{\mathrm{all}}$, the family of all measurable functions, all minimizability gap terms in Theorem 4.1 vanish, yielding the following result.

**Corollary 4.2.** *Fix a margin-based loss function $\Phi$. Assume that there exists a function $\Gamma(t) = \beta t^\alpha$ for some $\alpha \in (0, 1]$ and $\beta > 0$, such that the following excess error bound holds for all $h \in \mathcal{H}_{\mathrm{all}}$ and all distributions:*

$$\mathcal{E}_{\ell_{0-1}}(h) - \mathcal{E}_{\ell_{0-1}}^*(\mathcal{H}_{\mathrm{all}}) \leq \Gamma\big(\mathcal{E}_\Phi(h) - \mathcal{E}_\Phi^*(\mathcal{H}_{\mathrm{all}})\big).$$

*Then, the following excess error bound holds for all $h \in \mathcal{H}_{\mathrm{all}}$ and all distributions:*

$$\mathcal{E}_{\mathsf{L}}(h) - \mathcal{E}_{\mathsf{L}}^*(\mathcal{H}_{\mathrm{all}}) \leq \overline{\Gamma}\big(\mathcal{E}_{\mathsf{L}_\Phi}(h) - \mathcal{E}_{\mathsf{L}_\Phi}^*(\mathcal{H}_{\mathrm{all}})\big),$$

*where $\overline{\Gamma}(t) = \beta(2\,\mathsf{L}_{\max})^{1-\alpha}t^\alpha$.*

Building on the results of Awasthi et al. (2022a) for $\Gamma(t)$ already mentioned and Corollary 4.2, we now derive the following result.

**Corollary 4.3.** *For all $h \in \mathcal{H}_{\mathrm{all}}$ and all distributions,*

$$\mathcal{E}_{\mathsf{L}}(h) - \mathcal{E}_{\mathsf{L}}^*(\mathcal{H}_{\mathrm{all}}) \leq \overline{\Gamma}\big(\mathcal{E}_{\mathsf{L}_\Phi}(h) - \mathcal{E}_{\mathsf{L}_\Phi}^*(\mathcal{H}_{\mathrm{all}})\big),$$

*where $\overline{\Gamma}(t) = 2\sqrt{\mathsf{L}_{\max}\,t}$ for $\Phi = \Phi_{\exp}$ and $\Phi_{\log}$, $\overline{\Gamma}(t) = \sqrt{2\,\mathsf{L}_{\max}\,t}$ for $\Phi = \Phi_{\mathrm{quad}}$, and $\overline{\Gamma}(t) = t$ for $\Phi = \Phi_{\mathrm{hinge}}$, $\Phi_{\mathrm{sig}}$, and $\Phi_\rho$.*

By taking the limit on both sides, we establish the Bayes-consistency of these cost-sensitive surrogate losses $\mathsf{L}_\Phi$ with respect to the cost-sensitive target loss $\mathsf{L}$. More generally, Corollary 4.2 demonstrates that $\mathsf{L}_\Phi$ admits an excess error bound with respect to $\mathsf{L}$ if $\Phi$ admits an excess error bound with respect to $\ell_{0-1}$.

# 5. Algorithm for Generalized Metrics

In this section, we build on the previous theoretical analysis to develop algorithms for optimizing general metrics with strong guarantees. We first characterize $\lambda^*$, motivating a binary search algorithm under oracle access to the sign of the expected loss. We then propose an algorithm based on empirical minimization of a surrogate loss $\mathsf{L}_\Phi$ for $\ell^\lambda$ and introduce a simpler cross-validation approach for selecting $\lambda$. Finally, we discuss the theoretical foundations of our algorithms, compare them to prior work, and highlight cases where existing methods may fail due to reliance on the Bayes-optimal solution.

First, note that in the general cost-sensitive learning problem, the costs $\overline{\mathsf{L}}(+1, y)$ and $\overline{\mathsf{L}}(-1, y)$ are known *a priori*. In our scenario, the costs defining $\ell^{\lambda^*}$ depend on $\lambda^*$, which is not known. We will seek to determine or approximate $\lambda^*$. Recall that $\lambda^* = \mathcal{L}^*(\mathcal{H}) = \inf_{h \in \mathcal{H}} \frac{\mathbb{E}_{(x,y)\sim\mathcal{D}}[\ell_{\boldsymbol{\alpha}}(h,x,y)]}{\mathbb{E}_{(x,y)\sim\mathcal{D}}[\ell_{\boldsymbol{\beta}}(h,x,y)]}$ and that the expected loss of $h \in \mathcal{H}$ with respect to the loss function $\ell^\lambda$ can be expressed as follows:

$$\mathcal{E}_{\ell^\lambda}(h) = \mathbb{E}_{(x,y)\sim\mathcal{D}}[\ell_{\boldsymbol{\alpha}}(h,x,y)] - \lambda \mathbb{E}_{(x,y)\sim\mathcal{D}}[\ell_{\boldsymbol{\beta}}(h,x,y)].$$

The following provides a key characterization of $\lambda^*$.

**Theorem 5.1.** *We have $\mathcal{E}_{\ell^{\lambda^*}}^*(\mathcal{H}) = 0$ and, for any $\lambda \in \mathbb{R}$, $\mathrm{sign}\big(\mathcal{E}_{\ell^\lambda}^*(\mathcal{H})\big) = \mathrm{sign}(\lambda^* - \lambda)$.*

The proof is included in Appendix D. Theorem 5.1 provides a characterization of the sign of the best-in-class expected loss $\mathcal{E}_{\ell^\lambda}^*(\mathcal{H})$ in terms of the sign of $\lambda^* - \lambda$.

---

**Algorithm 1** Binary search estimation of $\lambda^*$

**input** $\epsilon$
1: Initialize $[a, b] \leftarrow [\lambda_{\min}, \lambda_{\max}]$
2: **repeat**
3:     $\lambda \leftarrow \frac{a+b}{2}$
4:     **if** $(\mathcal{E}_{\ell^\lambda}^*(\mathcal{H}) > 0)$ **then**
5:         $[a, b] = [\lambda, b]$
6:     **else**
7:         $[a, b] = [a, \lambda]$
8:     **end if**
9: **until** $|b - a| \leq \epsilon$
10: **return** $\lambda$

---

This naturally suggests a binary search-based algorithm to compute an $\epsilon$-approximation of $\lambda^*$, assuming oracle access to the sign of $\mathcal{E}_{\ell^\lambda}^*(\mathcal{H})$. The pseudocode of this algorithm is provided in Algorithm 1, where $\lambda_{\min}$ and $\lambda_{\max}$ denote the minimum and maximum possible values of $\lambda$, respectively. These bounds can be determined from the range of $\lambda^*$ using the formulation (2) for the given pair $(\boldsymbol{\alpha}, \boldsymbol{\beta})$.

**Theorem 5.2.** *Let $\epsilon > 0$ be fixed. Algorithm 1 returns an $\epsilon$-approximation $\lambda$ of $\lambda^*$ in $O\big(\log_2\big(\frac{\lambda_{\max} - \lambda_{\min}}{\epsilon}\big)\big)$ time, such that the following property holds:*

$$\mathcal{E}_{\ell^\lambda}^*(\mathcal{H}) \leq \mathcal{E}_{\ell^{\lambda^*}}^*(\mathcal{H}) + \epsilon\bar{\ell}_\beta = \epsilon\bar{\ell}_\beta.$$

The proof is presented in Appendix E. Of course, in practice, we do not have oracle access to the sign of $\mathcal{E}_{\ell^\lambda}^*(\mathcal{H})$. However, we can approximate $\mathcal{E}_{\ell^\lambda}^*(\mathcal{H})$ by computing the solution $\widehat{h}_S$ that minimizes a surrogate loss $\mathsf{L}_\Phi$ of $\ell^\lambda$ on a labeled sample $S$ of size $m$. To do that, we first provide a generalization bound for the target loss $\ell^\lambda$ by using our $\mathcal{H}$-consistency bounds presented in Section 4.3. Given a sample $S$ of size $m$, we denote by $\mathfrak{R}_m^\lambda(\mathcal{H})$ the Rademacher complexity of the function class $\{(x,y) \mapsto \mathsf{L}_\Phi(h,x,y) : h \in \mathcal{H}\}$ and $B_\lambda = \sup_{h,x,y} \mathsf{L}_\Phi(h,x,y)$ an upper bound on $\mathsf{L}_\Phi$.

**Theorem 5.3.** *Assume that the surrogate loss $\mathsf{L}_\Phi$ admits a $\overline{\Gamma}$-$\mathcal{H}$-consistency bound with respect to $\ell^\lambda$. Then, for any $\delta > 0$, with probability at least $1 - \delta$ over the draw of a sample $S$ from $\mathcal{D}^m$, the following estimation bound holds for an empirical minimizer $\widehat{h}_S \in \mathcal{H}$ of the $\mathsf{L}_\Phi$ over $S$:*

$$\mathcal{E}_{\ell^\lambda}(\widehat{h}_S) - \mathcal{E}_{\ell^\lambda}^*(\mathcal{H})$$
$$\leq \overline{\Gamma}\left(\mathcal{M}_{\mathsf{L}_\Phi}(\mathcal{H}) + 4\mathfrak{R}_m^\lambda(\mathcal{H}) + 2B_\lambda\sqrt{\frac{\log\frac{2}{\delta}}{2m}}\right) - \mathcal{M}_{\ell^\lambda}(\mathcal{H}).$$

Note that such $\ell^\lambda$-estimation loss guarantees for the minimizer of a surrogate loss $\mathsf{L}_\Phi$ can rarely be found in the literature. The proof is presented in Appendix F. For Theorem 5.3, the parameter $\lambda$ is fixed. But, the bound of the theorem can be generalized to hold uniformly for all $\lambda \in [\lambda_{\min}, \lambda_{\max}]$ by covering the interval using sub-intervals of size $1/m$.

**Theorem 5.4.** *Assume that the surrogate loss* $\mathsf{L}_\Phi$ *admits a* $\overline{\Gamma}$-$\mathcal{H}$-*consistency bound with respect to* $\ell^\lambda$. *Then, for any* $\delta > 0$, *with probability at least* $1 - \delta$ *over the draw of a sample $S$ from* $\mathcal{D}^m$, *the following estimation bound holds for an empirical minimizer* $\widehat{h}_S \in \mathcal{H}$ *of the surrogate loss* $\mathsf{L}_\Phi$ *over $S$ and* $\lambda \in [\lambda_{\min}, \lambda_{\max}]$:

$$\mathcal{E}_{\ell^\lambda}(\widehat{h}_S) - \mathcal{E}_{\ell^\lambda}^*(\mathcal{H}) \le \overline{\Gamma}\bigg(\mathcal{M}_{\mathsf{L}_\Phi}(\mathcal{H}) + 4\mathfrak{R}_m^\lambda(\mathcal{H})$$

$$+ \frac{8\max_i|\beta_i|B_\Phi}{m^2} + \bigg[2B_\lambda + \frac{4\max_i|\beta_i|B_\Phi}{m}\bigg]\sqrt{\frac{\log\frac{2\Delta\lambda m}{\delta}}{2m}}\bigg) + \frac{\overline{\ell}_\beta}{m},$$

*where* $\Delta\lambda = \lambda_{\max} - \lambda_{\min}$. *In particular, when* $\overline{\Gamma}(t) = 2(\mathsf{L}_{\max})^{\frac{1}{2}}t^{\frac{1}{2}}$ *for* $\Phi = \Phi_{\log}$, *the bound can be expressed as follows:*

$$\mathcal{E}_{\ell^\lambda}(\widehat{h}_S) - \mathcal{E}_{\ell^\lambda}^*(\mathcal{H}) \le (2\,\mathsf{L}_{\max})^{\frac{1}{2}}\bigg[\mathcal{M}_{\mathsf{L}_\Phi}(\mathcal{H}) + 4\mathfrak{R}_m^\lambda(\mathcal{H})$$

$$+ \frac{8\max_i|\beta_i|B_\Phi}{m^2} + \bigg[2B_\lambda + \frac{4\max_i|\beta_i|B_\Phi}{m}\bigg]\sqrt{\frac{\log\frac{2(\Delta\lambda)m}{\delta}}{2m}}\bigg]^{\frac{1}{2}} + \frac{\overline{\ell}_\beta}{m}.$$

Then, for any $\delta > 0$, with probability at least $1 - \delta$, for all $\lambda \in [\lambda_{\min}, \lambda_{\max}]$, we have

$$\mathcal{E}_{\ell^\lambda}(\widehat{h}_S) - \mathcal{E}_{\ell^\lambda}^*(\mathcal{H})$$
$$\le O\bigg(\overline{\Gamma}\Big(\mathfrak{R}_m^\lambda(\mathcal{H}) + \sqrt{\tfrac{\log((\lambda_{\max}-\lambda_{\min})m/\delta)}{m}} + \mathcal{M}_{\mathsf{L}_\Phi}(\mathcal{H})\Big)\bigg).$$

The proof is presented in Appendix G. We denote the right-hand of this bound by $\epsilon_m$. Building on the ideas from Algorithm 1, we introduce the modified algorithm Algorithm 2. Note that, with high probability, $\widehat{\mathcal{E}}_{\ell^\lambda,S}(\widehat{h}_\lambda) > \epsilon_m$ implies $\mathcal{E}_{\ell^\lambda}^*(\mathcal{H}) > 0$ and, similarly, $\widehat{\mathcal{E}}_{\ell^\lambda,S}(\widehat{h}_\lambda) < -\epsilon_m$ implies $\mathcal{E}_{\ell^\lambda}^*(\mathcal{H}) < 0$. Thus, this follows conditions used in the previous algorithm. The algorithm benefits from the following guarantee.

**Theorem 5.5.** *Let* $\epsilon = \frac{\epsilon_m}{2\overline{\ell}_\beta}$. *For any* $\delta > 0$, *with probability at least* $1 - \delta$, *Algorithm 2 returns in* $O\big(\log_2\big(\frac{\lambda_{\max}-\lambda_{\min}}{\epsilon}\big)\big)$ *time a hypothesis* $\widehat{h}_\lambda$ *that admits the following guarantee:*

$$\mathcal{L}(\widehat{h}_\lambda) \le \mathcal{L}^*(\mathcal{H}) + \frac{4\epsilon_m}{\underline{\ell}_\beta}.$$

The proof can be found in Appendix H. Thus, when $\epsilon_m$ is small, that is, when the sample size is sufficiently large relative to the complexity of $\mathcal{H}$ and the minimizability gap is small, the performance of $\widehat{h}_\lambda$ closely approaches the optimal performance achievable with $\mathcal{H}$.

In practice, the theoretical expression for $\epsilon_m$ may not be sufficiently tight due to constants or the minimizability gap, which cannot be accurately approximated in non-realizable

**Algorithm 2** Generalized metrics optimization algorithm

**input** $\epsilon, \epsilon_m$.
1: Initialize with $[a, b] = [\lambda_{\min}, \lambda_{\max}]$
2: **repeat**
3:     $\lambda \leftarrow \frac{a+b}{2}$
4:     $\widehat{h}_\lambda \leftarrow \operatorname{argmin}_{h\in\mathcal{H}} \widehat{\mathcal{E}}_{\mathsf{L}_\Phi,S}(h)$
5:     **if** $(\widehat{\mathcal{E}}_{\ell^\lambda,S}(\widehat{h}_\lambda) > \epsilon_m)$ **then**
6:         $[a, b] = [\lambda, b]$
7:     **else if** $(\widehat{\mathcal{E}}_{\ell^\lambda,S}(\widehat{h}_\lambda) < -\epsilon_m)$ **then**
8:         $[a, b] = [a, \lambda]$
9:     **else**
10:         **return** $\widehat{h}_\lambda$
11:     **end if**
12: **until** $|b - a| \le \epsilon$
13: **return** $\widehat{h}_\lambda$

**Algorithm 3** Generalized metrics optimization algorithm with cross-validation

**input** $\epsilon$
1: Initialize with $[a, b] = [\lambda_{\min}, \lambda_{\max}]$, $\lambda^* = \lambda_{\max}$, $i = 0$
2: **repeat**
3:     $\lambda \leftarrow a + i\epsilon$
4:     $\widehat{h}_\lambda \leftarrow \operatorname{argmin}_{h\in\mathcal{H}} \widehat{\mathcal{E}}_{\mathsf{L}_\Phi,S}(h)$
5:     **if** $(\widehat{\mathcal{L}}_S(\widehat{h}_\lambda) < \lambda^*)$ **then**
6:         $\widehat{\lambda} = \lambda$
7:         $\lambda^* \leftarrow \widehat{\mathcal{L}}_S(\widehat{h}_\lambda)$
8:     **end if**
9:     $i \leftarrow i + 1$
10: **until** $a + i\epsilon > b$
11: **return** $\widehat{h}_\lambda$

cases. In such scenarios, $\lambda$ can be treated as a hyperparameter and tuned via cross-validation, as outlined in Algorithm 3. The following result establishes convergence and performance guarantees for this algorithm. Compared to Algorithm 2, its computational complexity is linear rather than logarithmic.

**Theorem 5.6.** *For any* $\delta > 0$, *with probability at least* $1 - \delta$, *for* $\epsilon \le \frac{\epsilon_m}{2\overline{\ell}_\beta}$, *Algorithm 3 returns in* $O\big(\frac{\lambda_{\max}-\lambda_{\min}}{\epsilon}\big)$ *time a hypothesis* $\widehat{h}_\lambda$ *that admits the following guarantee:*

$$\mathcal{L}(\widehat{h}_\lambda) \le \mathcal{L}^*(\mathcal{H}) + \frac{2\epsilon_m}{\underline{\ell}_\beta}.$$

The proof is presented in Appendix I. We refer to Algorithms 2 and 3 as **METRO** (*Metric Optimization*). The effectiveness of **METRO** compared to prior baselines is demonstrated by the experimental results reported in Section 6.

Note that the quantities $\widehat{h}_\lambda$ and $\widehat{\mathcal{E}}_{\ell^\lambda,S}(\widehat{h}_\lambda)$ in Algorithm 2, as well as $\widehat{h}_\lambda$ and $\widehat{\mathcal{L}}_S(\widehat{h}_\lambda)$ in Algorithm 3, are approximated using data sampled from the same distribution, but they can be obtained from different samples. In practice, as done in

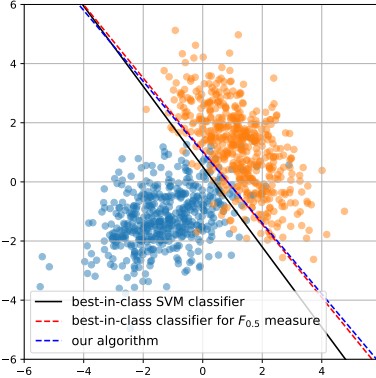

*Figure 1.* Comparison of the best-in-class SVM classifier, the optimal linear hypothesis for the $F_{0.5}$-measure, and the linear hypothesis returned by our algorithm.

(Koyejo et al., 2014), we can split the training data into two parts: $\widehat{\lambda}$ is obtained from one part, and then used to train the hypothesis $\widehat{h}_{\widehat{\lambda}}$ on the other.

Theorems 5.5 and 5.6 remain valid as long as the data are sampled independently from the same distribution. However, in over-parameterized settings, the value of $\epsilon_m$ may be larger due to the high complexity of the model. This quantity becomes small only when the sample size is sufficiently large relative to the complexity of the hypothesis set. This limitation applies broadly to most generalization bounds for complex neural networks.

The current analysis of over-parameterized settings typically requires alternative tools, particularly those that account for the optimization algorithm (e.g., Stochastic Gradient Descent (SGD) (Bottou, 2010)) and its dynamics. Such analyses often apply only to more restricted model families.

**Comparison of our algorithms with prior work.** As discussed in previous sections, our algorithms METRO for optimizing general metrics are supported by strong theoretical guarantees. These guarantees apply to arbitrary hypothesis sets and provide finite-sample bounds. In contrast, prior methods (Koyejo et al., 2014) rely solely on Bayes-consistency, which holds only for the class of all measurable functions. This makes their analysis less relevant in realistic settings where learning is restricted to specific function classes, an issue explicitly left open in (Koyejo et al., 2014). Furthermore, their approach does not provide convergence rate guarantees, in contrast with our finite-sample bounds. In many modern applications, particularly with complex neural networks, the minimizability gap is often small or close to zero due to near-separable data. In such cases, our learning guarantees become even more favorable.

Beyond theoretical advantages, a key limitation of prior methods is their reliance on the structure of the Bayes-optimal solution. Since the Bayes-optimal predictor for a given metric typically differs from that of binary classifi-

cation only by an offset, their approach first trains a binary classifier and then selects an optimal threshold or offset. However, this approach typically fails to find the best predictor within a restricted hypothesis set.

Figure 1 illustrates this issue with a simulated example. The best-in-class linear hypothesis for $\mathcal{L}$ with $\boldsymbol{\alpha} = -\left(\frac{5}{16}, \frac{5}{16}, \frac{5}{16}, \frac{5}{16}\right)$ and $\boldsymbol{\beta} = \left(0, \frac{1}{8}, \frac{1}{2}, \frac{5}{8}\right)$ (corresponding to the $F_{0.5}$-measure) is significantly different from the linear hypothesis obtained by thresholding the best-in-class SVM classifier. The two decision boundaries are not even parallel, highlighting the fundamental inadequacy of thresholding-based approaches such as those in (Koyejo et al., 2014). In contrast, our algorithm successfully finds a linear hypothesis that closely matches the best-in-class solution for $\mathcal{L}$.

Finally, note that when $\beta_1 = \beta_3 = 0$ for generalized metrics, the independence of $\lambda^*$ from the hypothesis $h$ in the target loss (Eq. (4)) significantly simplifies our algorithm, as $\lambda^*$ no longer needs to be estimated.

## 6. Experiments

In this section, we present empirical results for our principled algorithms for optimizing generalized metrics on the CIFAR-10 (Krizhevsky, 2009), CIFAR-100 (Krizhevsky, 2009) and SVHN (Netzer et al., 2011) datasets.

Our experiments use a three-hidden-layer CNN with ReLU activations (LeCun et al., 1995). Standard data augmentations were applied to CIFAR-10 and CIFAR-100, including 4-pixel padding followed by $32 \times 32$ random cropping and random horizontal flipping. Training was conducted using Stochastic Gradient Descent (SGD) with Nesterov momentum (Nesterov, 1983). The initial learning rate, batch size, and weight decay were set to $0.02$, $1,024$, and $1 \times 10^{-4}$, respectively. A cosine decay learning rate schedule (Loshchilov & Hutter, 2022) was used over the course of 100 epochs. During training, we extract two classes from each dataset to form a binary classification task.

We evaluated the models using their averaged generalized metric $\mathcal{L}$. In particular, we consider the $F_\beta$ measure (Ye et al., 2012), where $\boldsymbol{\alpha} = -\left(\frac{1+\beta^2}{4}, \frac{1+\beta^2}{4}, \frac{1+\beta^2}{4}, \frac{1+\beta^2}{4}\right)$ and $\boldsymbol{\beta} = \left(0, \frac{\beta^2}{2}, \frac{1}{2}, \frac{\beta^2+1}{2}\right)$, and the Jaccard similarity coefficient (JAC) (Sokolova & Lapalme, 2009), where $\boldsymbol{\alpha} = -\left(\frac{1}{4}, \frac{1}{4}, \frac{1}{4}, \frac{1}{4}\right)$ and $\boldsymbol{\beta} = \left(\frac{1}{4}, \frac{1}{4}, -\frac{1}{4}, \frac{3}{4}\right)$. The reported metric is averaged over five runs, with standard deviations included. We compared the METRO algorithm using cross-validation (Algorithm 3) with four baselines: the standard empirical risk minimization (ERM), the first algorithm (Algorithm 1) and the second algorithm (Algorithm 2) from (Koyejo et al., 2014), as well as the algorithm from (Parambath et al., 2014), as detailed in Section 1. For METRO algorithm, we used the surrogate loss $\mathsf{L}_\Phi$ in (6) with the auxiliary func-

*Table 2.* $F_\beta$ measure and JAC of three-hidden-layer neural network on CIFAR-10, CIFAR-100 and SVHN; mean ± standard deviation over five runs for ERM, Algorithm 1 and Algorithm 2 in (Koyejo et al., 2014), Algorithm in (Parambath et al., 2014), and METRO Algorithm.

| Algorithm | Metric | CIFAR-10 | CIFAR-100 | SVHN |
|---|---|---|---|---|
| ERM | | $0.9004 \pm 0.0015$ | $0.9265 \pm 0.0067$ | $0.9676 \pm 0.0028$ |
| Algorithm 1 in (Koyejo et al., 2014) | | $0.9040 \pm 0.0134$ | $0.9320 \pm 0.0065$ | $0.9679 \pm 0.0035$ |
| Algorithm 2 in (Koyejo et al., 2014) | $F_1$ | $0.9090 \pm 0.0070$ | $0.9317 \pm 0.0114$ | $0.9677 \pm 0.0022$ |
| Algorithm in (Parambath et al., 2014) | | $0.9185 \pm 0.0029$ | $0.9343 \pm 0.0111$ | $0.9682 \pm 0.0021$ |
| **METRO Algorithm** | | $\mathbf{0.9359 \pm 0.0041}$ | $\mathbf{0.9405 \pm 0.0103}$ | $\mathbf{0.9713 \pm 0.0029}$ |
| ERM | | $0.9418 \pm 0.0092$ | $0.9338 \pm 0.0216$ | $0.9689 \pm 0.0037$ |
| Algorithm 1 in (Koyejo et al., 2014) | | $0.9476 \pm 0.0032$ | $0.9510 \pm 0.0138$ | $0.9715 \pm 0.0035$ |
| Algorithm 2 in (Koyejo et al., 2014) | $F_{0.5}$ | $0.9503 \pm 0.0028$ | $0.9435 \pm 0.0140$ | $0.9730 \pm 0.0017$ |
| Algorithm in (Parambath et al., 2014) | | $0.9507 \pm 0.0036$ | $0.9515 \pm 0.0105$ | $0.9746 \pm 0.0011$ |
| **METRO Algorithm** | | $\mathbf{0.9585 \pm 0.0023}$ | $\mathbf{0.9759 \pm 0.0115}$ | $\mathbf{0.9807 \pm 0.0015}$ |
| ERM | | $0.9234 \pm 0.0074$ | $0.9279 \pm 0.0160$ | $0.9675 \pm 0.0035$ |
| Algorithm 1 in (Koyejo et al., 2014) | | $0.9305 \pm 0.0029$ | $0.9331 \pm 0.0218$ | $0.9688 \pm 0.0025$ |
| Algorithm 2 in (Koyejo et al., 2014) | $F_{1.5}$ | $0.9263 \pm 0.0036$ | $0.9340 \pm 0.0119$ | $0.9677 \pm 0.0020$ |
| Algorithm in (Parambath et al., 2014) | | $0.9312 \pm 0.0044$ | $0.9345 \pm 0.0118$ | $0.9702 \pm 0.0023$ |
| **METRO Algorithm** | | $\mathbf{0.9459 \pm 0.0030}$ | $\mathbf{0.9449 \pm 0.0115}$ | $\mathbf{0.9771 \pm 0.0018}$ |
| ERM | | $0.4689 \pm 0.0022$ | $0.4693 \pm 0.0063$ | $0.4767 \pm 0.0036$ |
| Algorithm 1 in (Koyejo et al., 2014) | | $0.4821 \pm 0.0024$ | $0.4728 \pm 0.0109$ | $0.4814 \pm 0.0031$ |
| Algorithm 2 in (Koyejo et al., 2014) | JAC | $0.5746 \pm 0.0121$ | $0.4753 \pm 0.0119$ | $0.4875 \pm 0.0017$ |
| Algorithm in (Parambath et al., 2014) | | $0.6428 \pm 0.0130$ | $0.4924 \pm 0.0152$ | $0.4934 \pm 0.0019$ |
| **METRO Algorithm** | | $\mathbf{0.6575 \pm 0.0018}$ | $\mathbf{0.5017 \pm 0.0046}$ | $\mathbf{0.4965 \pm 0.0024}$ |

tion $\Phi(t) = \log(1 + e^{-t})$, which corresponds to the logistic loss function used in logistic regression. The logistic loss was also used for ERM and Algorithm 1 in (Koyejo et al., 2014), while Algorithm 2 in (Koyejo et al., 2014) and the algorithm from (Parambath et al., 2014) used the weighted logistic loss. All the hyperparameters in these algorithms were selected through cross-validation.

Table 2 shows that our algorithm consistently outperforms the four baselines across all datasets. Notably, the algorithm in (Parambath et al., 2014) consistently outperforms Algorithms 1 and 2 in (Koyejo et al., 2014) by using two hyperparameters: $\theta$ for weighting the logistic loss and $\theta'$ for thresholding the classifier. In contrast, the relative performance of Algorithms 1 and 2 in (Koyejo et al., 2014) varies across datasets, although both outperform ERM.

Note that the per-epoch computational cost of our method is comparable to that of Algorithm 2 in (Koyejo et al., 2014). Both methods involve a single hyperparameter, and for a fixed value of this parameter, the computational cost is similar to training a standard binary classifier using a standard surrogate loss.

In line with prior work (Koyejo et al., 2014), we used standard image classification datasets for our empirical evaluation to demonstrate the effectiveness of our methods relative to existing baselines. However, we acknowledge the importance of evaluating our algorithms on more imbalanced

datasets, which often present greater challenges and are more representative of real-world applications. We plan to include such experiments and comparisons in future work.

One may wonder about the difficulty of directly optimizing a general metric defined as the ratio of the expectations of two loss functions, both linear in h, where $h(x) = \text{sign}(h(x))$. While the metric is quasi-concave in h, optimizing it with respect to $h$ is NP-hard, even when the denominator is constant and $h$ is restricted to a linear hypothesis set. In contrast, each surrogate loss optimization problem we consider (framed as supervised learning) can be solved in polynomial time over a convex hypothesis set, as the surrogate loss functions we adopt are convex. Furthermore, directly optimizing the empirical ratio of the numerator and denominator may not yield a provably good approximation of the metric, since their expectation does not align with the ratio of expectations.

## 7. Conclusion

We presented a series of theoretical, algorithmic, and empirical results for optimizing generalized metrics in binary classification, highlighting the significance of our algorithms supported by $\mathcal{H}$-consistency guarantees. Looking ahead, a natural direction is extending our theory and algorithms to cover generalized metrics in multi-class classification and multi-label learning, broadening their applicability.

## Acknowledgements

We thank the anonymous reviewers for their valuable feedback and constructive suggestions.

## Impact Statement

This paper presents work whose goal is to advance the field of Machine Learning. There are many potential societal consequences of our work, none which we feel must be specifically highlighted here.

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

# Contents of Appendix

# A. Related work

In binary classification, zero-one misclassification loss may not always serve as an appropriate evaluation metric, particularly in scenarios where more complex metrics are better suited to the problem. For instance, in scenarios with significant class imbalance, which frequently arise in applications such as fraud detection, medical diagnosis, and text retrieval (Lewis, 1995b; Drummond & Holte, 2005; He & Garcia, 2009; Gu et al., 2009), metrics like the $F_\beta$-*measure* (Lewis, 1995a; Jansche, 2005; Ye et al., 2012) and the *AM measure* (Menon et al., 2013) are commonly used. Similarly, *weighted accuracy* is often used to address the asymmetrical costs associated with different classes in real-world applications. However, optimizing these generalized performance metrics introduces both computational and statistical challenges, as they do not conform to the standard single-loss expectations commonly used for analyzing surrogate loss functions in machine learning (Steinwart, 2007).

Research on the design of algorithms for specific instances of such metrics or the general family has largely focused on the characterization of the Bayes-optimal classifier. For the standard zero-one binary classification loss, it is known that the Bayes classifier can be defined as $x \mapsto \text{sign}(\eta(x) - \frac{1}{2})$, where $\eta(x)$ is the probability of a positive label conditioned on $x$. A similar characterization applies to the weighted zero-one loss (Scott, 2012), where the threshold corresponds to a weight different from $\frac{1}{2}$. For imbalanced metrics, Ye et al. (2012) characterized the Bayes-classifier for the $F_1$-measure, while Menon et al. (2013) characterized the Bayes classifier for the AM measure as $x \mapsto \text{sign}(\eta(x) - \delta^*)$ for some $\delta^* \in (0, 1)$. These results were later generalized by Koyejo et al. (2014) to a broader family of metrics that can be formulated as a ratio of two linear functions of true positive (TP), false positive (FP), true negative (TN), and false negative (FN) statistics. This family of linear-fractional metrics includes the aforementioned weighted accuracy, $F_\beta$-measures, the AM measure, as well as the *Jaccard similarity coefficient* (Sokolova & Lapalme, 2009), among others. Existing algorithms for optimizing these metrics heavily rely on the structure of the Bayes-optimal solution (Koyejo et al., 2014; Parambath et al., 2014). These methods generally adopt a two-stage approach: first estimating the conditional probability $\eta(x)$ and then searching for a suitable threshold $\delta^*$ specific to the metric.

In particular, Koyejo et al. (2014) study a general family of performance metrics, including the $F_\beta$-measure, AM measure, Jaccard similarity coefficient (JAC), and Weighted Accuracy (WA). These metrics can be expressed as the ratio of two linear combinations of four fundamental classification quantities. The authors propose two relatively simple algorithms. Their first algorithm consists of training a standard binary classifier such as logistic regression to return a real-valued predictor $h$. Next, a threshold $\theta$ is chosen to maximize the empirical $L$-measure for binary classifier $x \mapsto \text{sign}(h(x) - \theta)$, where $L$ represents the metric of interest. The authors do not discuss how to find $\theta$ but presumably this can be done via cross-validation based on a grid, or by binary search over all possible values. Their second algorithm consists of training for each fixed value of $\theta$ a cost-sensitive logistic regression (or other margin-based algorithm) with weights $\theta$ and $(1 - \theta)$ and return $h_\theta$. Then, they find $\theta$ to maximize the empirical $L$-measure for the binary classifier $x \mapsto \text{sign}(h_\theta(x))$. The authors provide a consistency-type guarantee for these two-stage algorithms. Parambath et al. (2014) address the specific case of the $F_1$-measure. Their algorithm coincides with the second algorithm of Koyejo et al. (2014). However, the authors also suggest returning $\text{sign}(h_\theta(x) - \theta')$ where both $\theta$ and $\theta'$ are selected to maximize the $L$-measure. The authors provide a stability-type guarantee for their method, although the analysis lacks explicit details.

These algorithms naturally come with consistency guarantees (Zhang, 2004; Bartlett et al., 2006; Steinwart, 2007; Koyejo et al., 2014; Mohri et al., 2018). However, consistency is an asymptotic property and does not provide explicit convergence rate guarantees. Moreover, it applies only to the overly broad class of all measurable functions, making it less relevant in practical scenarios where learning is constrained to a restricted hypothesis class (Long & Servedio, 2013; Zhang & Agarwal, 2020; Awasthi et al., 2021a;b; 2023; 2024; Mao et al., 2023d;c;a;b;e; Zheng et al., 2023; Mao et al., 2024a;b;c;h;e;d;g;f; Mohri et al., 2024; Cortes et al., 2024; 2025; Mao et al., 2025; Mao, 2025; Montreuil et al., 2024; 2025d;a;b;c; Zhong, 2025).

Other related work on generalized metrics includes studies on surrogate regret bounds (Reid & Williamson, 2009; Kotlowski & Dembczyński, 2016), extensions of the plug-in rule (Ye et al., 2012; Dembczynski et al., 2013; Narasimhan et al., 2014; Lipton et al., 2014; Dembczyński et al., 2017; Yan et al., 2018; Tavker et al., 2020; Berger & Guda, 2020), and structural loss optimization (Joachims, 2005; Kar et al., 2014; Yu & Blaschko, 2015; Eban et al., 2017; Berman et al., 2018; Bao & Sugiyama, 2020). Additionally, various optimization approaches have been explored, including online optimization (Kar et al., 2014; Zhang et al., 2018; Kotłowski et al., 2024; Busa-Fekete et al., 2015) and constrained optimization (Narasimhan et al., 2019). Further extensions address multi-class classification and multi-label learning (Dembczynski et al., 2013; Narasimhan et al., 2015a; Ramaswamy et al., 2015; Narasimhan et al., 2015b; 2016; Cheng et al., 2016; Natarajan et al.,

2016; Sanyal et al., 2018; Fathony & Kolter, 2020; Zhang et al., 2020; Luo et al., 2021; Busa-Fekete et al., 2022; Schultheis et al., 2024), as well as generalized metrics under missing, corrupted, or noisy labels (Menon et al., 2015; Natarajan & Jain, 2016; Zhang et al., 2021; Zhang & Agarwal, 2024). Theoretical analyses and algorithms have also been developed for specific metrics, including the $F$-measure (Jansche, 2007; Jasinska et al., 2016; Pillai et al., 2017; Bascol et al., 2019; Jiang et al., 2020; Berger & Guda, 2020; Zhang et al., 2020; Dai & Li, 2023), fairness measures (Menon & Williamson, 2018), precision-recall (Flach & Kull, 2015), and the balanced error rate (BER) (Zhao et al., 2013).

## B. Proof of Theorem 3.2

**Theorem 3.2.** *Fix $\eta \geq 0$ and $h \in \mathcal{H}$. Then, the inequality $\mathcal{E}_{\ell^{\lambda^*}}(h) \leq \eta$ holds if and only if $\mathcal{L}(h) - \mathcal{L}^*(\mathcal{H}) \leq \frac{\eta}{\mathbb{E}_{(x,y)\sim\mathcal{D}}[\ell_{\boldsymbol{\beta}}(h,x,y)]}$.*

*Proof.* Since we have $\mathbb{E}_{(x,y)\sim\mathcal{D}}[\ell_{\boldsymbol{\beta}}(h,x,y)] > 0$, the following equivalence holds for all $h \in \mathcal{H}$ and $\eta \geq 0$:

$$\mathcal{E}_{\ell^{\lambda^*}}(h) \leq \eta \iff \mathop{\mathbb{E}}_{(x,y)\sim\mathcal{D}}[\ell_{\boldsymbol{\alpha}}(h,x,y)] - \lambda^* \mathop{\mathbb{E}}_{(x,y)\sim\mathcal{D}}[\ell_{\boldsymbol{\beta}}(h,x,y)] \leq \eta \qquad \text{(def. of } \ell^{\lambda^*})$$

$$\iff \frac{\mathbb{E}_{(x,y)\sim\mathcal{D}}[\ell_{\boldsymbol{\alpha}}(h,x,y)]}{\mathbb{E}_{(x,y)\sim\mathcal{D}}[\ell_{\boldsymbol{\beta}}(h,x,y)]} \leq \lambda^* + \frac{\eta}{\mathbb{E}_{(x,y)\sim\mathcal{D}}[\ell_{\boldsymbol{\beta}}(h,x,y)]} \qquad (\mathbb{E}_{(x,y)\sim\mathcal{D}}[\ell_{\boldsymbol{\beta}}(h,x,y)] > 0)$$

$$\iff \mathcal{L}(h) - \mathcal{L}^*(\mathcal{H}) \leq \frac{\eta}{\mathbb{E}_{(x,y)\sim\mathcal{D}}[\ell_{\boldsymbol{\beta}}(h,x,y)]}. \qquad (\lambda^* = \mathcal{L}^*(\mathcal{H}))$$

This completes the proof. $\qquad\square$

## C. Proof of Theorem 4.1

We will use the following definitions. For any $x \in \mathcal{X}$, we adopt the definition $\eta(x) = \mathbb{P}(Y = +1 \mid X = x)$. Then, the *conditional loss* of a hypothesis $h$ at point $x \in \mathcal{X}$ for a loss function $\ell$ is defined as follows:

$$\mathcal{C}_\ell(h,x) = \eta(x)\ell(h,x,+1) + (1 - \eta(x))\ell(h,x,-1).$$

The *best-in-class conditional loss* of a hypothesis set $\mathcal{H}$ at $x \in \mathcal{X}$ is defined by $\mathcal{C}_\ell^*(\mathcal{H},x) = \inf_{h\in\mathcal{H}} \mathcal{C}_\ell(h,x)$ .

**Lemma C.1.** *Assume that the following $\mathcal{H}$-consistency bound holds for all $h \in \mathcal{H}$ and all distributions:*

$$\mathcal{E}_{\ell_{0-1}}(h) - \mathcal{E}_{\ell_{0-1}}^*(\mathcal{H}) + \mathcal{M}_{\ell_{0-1}}(\mathcal{H}) \leq \Gamma(\mathcal{E}_\Phi(h) - \mathcal{E}_\Phi^*(\mathcal{H}) + \mathcal{M}_\Phi(\mathcal{H})).$$

*Then, for any $\eta \in [0,1]$ and $x \in \mathcal{X}$, we have*

$$\ell_{0-1}(h,x,+1)\eta + \ell_{0-1}(h,x,-1)(1-\eta) - \inf_{h\in\mathcal{H}}(\ell_{0-1}(h,x,+1)\eta + \ell_{0-1}(h,x,-1)(1-\eta))$$

$$\leq \Gamma\Big(\Phi(h(x))\eta + \Phi(-h(x))(1-\eta) - \inf_{h\in\mathcal{H}}(\Phi(h(x))\eta + \Phi(-h(x))(1-\eta))\Big).$$

*Proof.* For any $x \in \mathcal{X}$, consider a distribution $\delta_x$ that concentrates on that point. Let $\eta = \mathbb{P}(Y = +1 \mid X = x)$. Then, by definition, $\mathcal{E}_{\ell_{0-1}}(h) - \mathcal{E}_{\ell_{0-1}}^*(\mathcal{H}) + \mathcal{M}_{\ell_{0-1}}(\mathcal{H})$ can be expressed as

$$\mathcal{E}_{\ell_{0-1}}(h) - \mathcal{E}_{\ell_{0-1}}^*(\mathcal{H}) + \mathcal{M}_{\ell_{0-1}}(\mathcal{H}) = \ell_{0-1}(h,x,+1)\eta + \ell_{0-1}(h,x,-1)(1-\eta) - \inf_{h\in\mathcal{H}}(\ell_{0-1}(h,x,+1)\eta + \ell_{0-1}(h,x,-1)(1-\eta)).$$

Similarly, $\mathcal{E}_\Phi(h) - \mathcal{E}_\Phi^*(\mathcal{H}) + \mathcal{M}_\Phi(\mathcal{H})$ can be expressed as

$$\Phi(h(x))\eta + \Phi(-h(x))(1-\eta) - \inf_{h\in\mathcal{H}}(\Phi(h(x))\eta + \Phi(-h(x))(1-\eta)).$$

Since the $\mathcal{H}$-consistency bound holds by the assumption, we complete the proof. $\qquad\square$

**Theorem 4.1.** *Assume that $\mathsf{L}$ takes values in $[0, \mathsf{L}_{\max}]$. Let $\mathcal{H}$ be a regular hypothesis set and $\Phi$ a margin-based loss function for the binary zero-one loss function $\ell_{0-1}$. Assume that $\Phi$ admits a $\Gamma$-$\mathcal{H}$-consistency bound with respect to $\ell_{0-1}$ for a function $\Gamma: t \mapsto \beta\, t^\alpha$, with $\alpha \in (0,1]$ and $\beta > 0$. Then, $\mathsf{L}_\Phi$ admits a $\overline{\Gamma}$-$\mathcal{H}$-consistency bound with respect to $\mathsf{L}$, where $\overline{\Gamma}(t) = \beta(2\,\mathsf{L}_{\max})^{1-\alpha}t^\alpha$.*

*Proof.* Let $\eta(x) = \mathbb{P}(Y = 1 | X = x)$ be the conditional probability of $Y = 1$ given $X = x$. By the definition, the conditional loss of the target loss can be expressed as follows:

$$\mathcal{C}_{\mathsf{L}}(h, x) = \eta(x)\mathsf{L}(\mathsf{h}(x), +1) + (1 - \eta(x))\mathsf{L}(\mathsf{h}(x), -1)$$

$$= 1_{\mathsf{h}(x)=+1}[\eta(x)\mathsf{L}(+1, +1) + (1 - \eta(x))\mathsf{L}(+1, -1)] + 1_{\mathsf{h}(x)=-1}[\eta(x)\mathsf{L}(-1, +1) + (1 - \eta(x))\mathsf{L}(-1, -1)]$$

$$= [\ell_{0-1}(h, x, +1)\eta'(x) + \ell_{0-1}(h, x, -1)(1 - \eta'(x))]c(x),$$

where $c(x) = [\eta(x)\mathsf{L}(+1, +1) + (1 - \eta(x))\mathsf{L}(+1, -1)] + [\eta(x)\mathsf{L}(-1, +1) + (1 - \eta(x))\mathsf{L}(-1, -1)] \in [0, 2\mathsf{L}_{\max}]$ and $\eta'(x) = \frac{\eta(x)\mathsf{L}(-1,+1)+(1-\eta(x))\mathsf{L}(-1,-1)}{c(x)} \in [0, 1]$. Thus, the best-in-class conditional loss can be expressed as follows:

$$\mathcal{C}_{\mathsf{L}}^*(\mathcal{H}, x) = \inf_{h \in \mathcal{H}}[\ell_{0-1}(h, x, -1)\eta'(x) + \ell_{0-1}(h, x, -1)(1 - \eta'(x))]c(x)$$

By the definition,

$$\Delta\mathcal{C}_{\mathsf{L},\mathcal{H}}(h, x)$$
$$= \mathcal{C}_{\mathsf{L}}(h, x) - \mathcal{C}_{\mathsf{L}}^*(\mathcal{H}, x)$$
$$= \Big([\ell_{0-1}(h, x, -1)\eta'(x) + \ell_{0-1}(h, x, -1)(1 - \eta'(x))] - \inf_{h \in \mathcal{H}}[\ell_{0-1}(h, x, -1)\eta'(x) + \ell_{0-1}(h, x, -1)(1 - \eta'(x))]\Big)c(x).$$

The conditional loss of the surrogate loss can be expressed as follows:

$$\mathcal{C}_{\mathsf{L}_\Phi}(h, x) = \eta(x)(\mathsf{L}(+1, +1)\Phi(-h(x)) + \mathsf{L}(-1, +1)\Phi(h(x))) + (1 - \eta(x))(\mathsf{L}(+1, -1)\Phi(-h(x)) + \mathsf{L}(-1, -1)\Phi(h(x)))$$

$$= \Phi(-h(x))[\eta(x)\mathsf{L}(+1, +1) + (1 - \eta(x))\mathsf{L}(+1, -1)] + \Phi(h(x))[\eta(x)\mathsf{L}(-1, +1) + (1 - \eta(x))\mathsf{L}(-1, -1)]$$

$$= [\Phi(h(x))\eta'(x) + \Phi(-h(x))(1 - \eta'(x))]c(x).$$

Thus, the best-in-class conditional loss can be expressed as follows:

$$\mathcal{C}_{\mathsf{L}_\Phi}^*(\mathcal{H}, x) = \inf_{h \in \mathcal{H}}[\Phi(h(x))\eta'(x) + \Phi(-h(x))(1 - \eta'(x))]c(x).$$

By the definition,

$$\Delta\mathcal{C}_{\mathsf{L}_\Phi,\mathcal{H}}(h, x)$$
$$= \mathcal{C}_{\mathsf{L}_\Phi}(h, x) - \mathcal{C}_{\mathsf{L}_\Phi}^*(\mathcal{H}, x)$$
$$= \Big([\Phi(h(x))\eta'(x) + \Phi(-h(x))(1 - \eta'(x))] - \inf_{h \in \mathcal{H}}[\Phi(h(x))\eta'(x) + \Phi(-h(x))(1 - \eta'(x))]\Big)c(x).$$

By Lemma C.1, we have

$$\Delta\mathcal{C}_{\mathsf{L}_\Phi,\mathcal{H}}(h, x)$$

$$\Big([\Phi(h(x))\eta'(x) + \Phi(-h(x))(1 - \eta'(x))] - \inf_{h \in \mathcal{H}}[\Phi(h(x))\eta'(x) + \Phi(-h(x))(1 - \eta'(x))]\Big)c(x)$$

$$\geq \Gamma^{-1}\Big([\ell_{0-1}(h, x, -1)\eta'(x) + \ell_{0-1}(h, x, -1)(1 - \eta'(x))] - \inf_{h \in \mathcal{H}}[\ell_{0-1}(h, x, -1)\eta'(x) + \ell_{0-1}(h, x, -1)(1 - \eta'(x))]\Big)c(x)$$

$$= \frac{1}{\beta^{\frac{1}{\alpha}}}\Big([\ell_{0-1}(h, x, -1)\eta'(x) + \ell_{0-1}(h, x, -1)(1 - \eta'(x))] - \inf_{h \in \mathcal{H}}[\ell_{0-1}(h, x, -1)\eta'(x) + \ell_{0-1}(h, x, -1)(1 - \eta'(x))]\Big)^{\frac{1}{\alpha}}c(x)$$

$$= \frac{1}{\beta^{\frac{1}{\alpha}}}\Big(([\ell_{0-1}(h, x, -1)\eta'(x) + \ell_{0-1}(h, x, -1)(1 - \eta'(x))]$$

$$- \inf_{h \in \mathcal{H}}[\ell_{0-1}(h, x, -1)\eta'(x) + \ell_{0-1}(h, x, -1)(1 - \eta'(x))])c(x)\Big)^{\frac{1}{\alpha}}(c(x))^{1-\frac{1}{\alpha}}$$

$$\geq \frac{1}{\beta^{\frac{1}{\alpha}}}(\Delta\mathcal{C}_{\mathsf{L},\mathcal{H}}(h, x))^{\frac{1}{\alpha}}(2\mathsf{L}_{\max})^{1-\frac{1}{\alpha}}.$$

By taking the expectation on both sides and applying Jensen's inequality, we obtain:

$$\mathcal{E}_{\mathsf{L}}(h) - \mathcal{E}_{\mathsf{L}}^*(\mathcal{H}) + \mathcal{M}_{\mathsf{L}}(\mathcal{H}) \leq \overline{\Gamma}\big(\mathcal{E}_{\mathsf{L}_\Phi}(h) - \mathcal{E}_{\mathsf{L}_\Phi}^*(\mathcal{H}) + \mathcal{M}_{\mathsf{L}_\Phi}(\mathcal{H})\big),$$

where $\overline{\Gamma}(t) = \beta(2\mathsf{L}_{\max})^{1-\alpha}t^\alpha$. $\qquad\square$

# D. Proof of Theorem 5.1

**Theorem 5.1.** *We have $\mathcal{E}_{\ell^{\lambda*}}^*(\mathcal{H}) = 0$ and, for any $\lambda \in \mathbb{R}$, $\mathrm{sign}\big(\mathcal{E}_{\ell^\lambda}^*(\mathcal{H})\big) = \mathrm{sign}(\lambda^* - \lambda)$.*

*Proof.* For any $h \in \mathcal{H}$, define $f(h)$ and $g(h)$ to simplify notation:

$$f(h) = \frac{\mathbb{E}_{(x,y)\sim\mathcal{D}}[\ell_{\boldsymbol{\alpha}}(h,x,y)]}{\mathbb{E}_{(x,y)\sim\mathcal{D}}[\ell_{\boldsymbol{\beta}}(h,x,y)]}, \quad g(h) = \mathbb{E}_{(x,y)\sim\mathcal{D}}[\ell_{\boldsymbol{\beta}}(h,x,y)].$$

By assumption, we have $g(h) > 0$ for all $h \in \mathcal{H}$ and $\lambda^* = \inf_{h\in\mathcal{H}} f(h)$ and $\mathcal{E}_{\ell^{\lambda*}}^*(\mathcal{H}) = \inf_{h\in\mathcal{H}}\{(f(h) - \lambda^*)g(h)\}$. Since $g$ is upper-bounded by $\overline{\ell}_\beta$ and $(f(h) - \lambda^*) \geq 0$, it follows that

$$\mathcal{E}_{\ell^{\lambda*}}^*(\mathcal{H}) \leq \inf_{h\in\mathcal{H}}\{f(h) - \lambda^*\}\,\overline{\ell}_\beta = 0.$$

By definition of $\mathcal{E}_{\ell^{\lambda*}}^*(\mathcal{H})$ as an infimum, for any $\eta > 0$, there exists $h_\eta \in \mathcal{H}$ such that

$$\mathcal{E}_{\ell^{\lambda*}}^*(\mathcal{H}) + \eta > (f(h_\eta) - \lambda^*)g(h_\eta) \geq 0.$$

Since $\mathcal{E}_{\ell^{\lambda*}}^*(\mathcal{H}) + \eta > 0$ for all $\eta > 0$, it follows that $\mathcal{E}_{\ell^{\lambda*}}^*(\mathcal{H}) \geq 0$. Combining the two inequalities yields $\mathcal{E}_{\ell^{\lambda*}}^*(\mathcal{H}) = 0$. This establishes the first equality.

Next, assume $\lambda^* - \lambda > 0$. By the definition of $\lambda^*$ as an infimum, we have $f(h) - \lambda > 0$ for all $h \in \mathcal{H}$. This implies $\mathcal{E}_{\ell^\lambda}^*(\mathcal{H}) = \inf_{h\in\mathcal{H}}\{(f(h) - \lambda)g(h)\} \geq \underline{\ell}_\beta \inf_{h\in\mathcal{H}}\{(f(h) - \lambda)\} = \underline{\ell}_\beta(\lambda^* - \lambda) > 0$, thus $\mathcal{E}_{\ell^\lambda}^*(\mathcal{H}) > 0$, which proves one direction of the statement.

Now, assume $\lambda^* - \lambda < 0$. By the definition of $\lambda^*$ as an infimum, for any $\eta > 0$, there exists $h_\eta \in \mathcal{H}$ such that $f(h_\eta) < \lambda^* + \eta$. Choose $\eta < (\lambda - \lambda^*)$. This implies $\mathcal{E}_{\ell^\lambda}^*(\mathcal{H}) \leq (f(h_\eta) - \lambda)g(h_\eta) \leq (\lambda^* + \eta - \lambda)g(h_\eta)$. Since $(\lambda^* + \eta - \lambda) < 0$ and $g(h_\eta) > 0$, it follows that $\mathcal{E}_{\ell^\lambda}^*(\mathcal{H}) < 0$. This proves the other direction. $\square$

# E. Proof of Theorem 5.2

**Lemma E.1.** *Fix $\epsilon > 0$ and assume $|\lambda - \lambda^*| \leq \epsilon$. Then, the following inequality holds:*

$$\mathcal{E}_{\ell^\lambda}^*(\mathcal{H}) \leq \epsilon\overline{\ell}_\beta.$$

*Proof.* By definition of $\mathcal{E}_{\ell^\lambda}(h)$, the following holds for any $h \in \mathcal{H}$:

$$\mathcal{E}_{\ell^\lambda}(h) = \mathcal{E}_{\ell^{\lambda*}}(h) + (\lambda^* - \lambda)\mathbb{E}_{(x,y)\sim\mathcal{D}}[\ell_\beta(h,x,y)] \leq \mathcal{E}_{\ell^{\lambda*}}(h) + \epsilon\overline{\ell}_\beta.$$

Thus, by definition of $\mathcal{E}_{\ell^\lambda}^*(\mathcal{H})$ as an infimum, for any $h \in \mathcal{H}$, we have $\mathcal{E}_{\ell^\lambda}^*(\mathcal{H}) \leq \mathcal{E}_{\ell^{\lambda*}}(h) + \epsilon\overline{\ell}_\beta$. Taking the infimum of the right-hand side over $\mathcal{H}$ yields $\mathcal{E}_{\ell^\lambda}^*(\mathcal{H}) \leq \mathcal{E}_{\ell^{\lambda*}}^*(\mathcal{H}) + \epsilon\overline{\ell}_\beta$. By Theorem 5.1, we have $\mathcal{E}_{\ell^{\lambda*}}^*(\mathcal{H}) = 0$. This completes the proof. $\square$

**Theorem 5.2.** *Let $\epsilon > 0$ be fixed. Algorithm 1 returns an $\epsilon$-approximation $\lambda$ of $\lambda^*$ in $O\big(\log_2\big(\frac{\lambda_{\max} - \lambda_{\min}}{\epsilon}\big)\big)$ time, such that the following property holds:*

$$\mathcal{E}_{\ell^\lambda}^*(\mathcal{H}) \leq \mathcal{E}_{\ell^{\lambda*}}^*(\mathcal{H}) + \epsilon\overline{\ell}_\beta = \epsilon\overline{\ell}_\beta.$$

*Proof.* By definition of the binary search, we have $|\lambda - \lambda^*| \leq \epsilon$. Thus, the inequality holds by Lemma E.1. The time complexity follows straightforwardly the property of the binary search. $\square$

# F. Proof of Theorem 5.3

**Theorem 5.3.** *Assume that the surrogate loss $\mathsf{L}_\Phi$ admits a $\overline{\Gamma}$-$\mathcal{H}$-consistency bound with respect to $\ell^\lambda$. Then, for any $\delta > 0$, with probability at least $1 - \delta$ over the draw of a sample $S$ from $\mathcal{D}^m$, the following estimation bound holds for an empirical minimizer $\widehat{h}_S \in \mathcal{H}$ of the $\mathsf{L}_\Phi$ over $S$:*

$$\mathcal{E}_{\ell^\lambda}(\widehat{h}_S) - \mathcal{E}_{\ell^\lambda}^*(\mathcal{H}) \leq \Gamma\left(\mathcal{M}_{\mathsf{L}_\Phi}(\mathcal{H}) + 4\mathfrak{R}_m^\lambda(\mathcal{H}) + 2B_\lambda\sqrt{\frac{\log\frac{2}{\delta}}{2m}}\right) - \mathcal{M}_{\ell^\lambda}(\mathcal{H}).$$

*Proof.* By the standard Rademacher complexity bounds (Mohri et al., 2018), the following holds with probability at least $1 - \delta$ for all $h \in \mathcal{H}$:

$$\left|\mathcal{E}_{\mathsf{L}_\Phi}(h) - \widehat{\mathcal{E}}_{\mathsf{L}_\Phi, S}(h)\right| \leq 2\mathfrak{R}_m^\lambda(\mathcal{H}) + B_\lambda\sqrt{\frac{\log(2/\delta)}{2m}}.$$

Fix $\epsilon > 0$. By the definition of the infimum, there exists $h^* \in \mathcal{H}$ such that $\mathcal{E}_{\mathsf{L}_\Phi}(h^*) \leq \mathcal{E}_{\mathsf{L}_\Phi}^*(\mathcal{H}) + \epsilon$. By definition of $\widehat{h}_\lambda$, we have

$$\begin{aligned}
&\mathcal{E}_{\mathsf{L}_\Phi}(\widehat{h}_\lambda) - \mathcal{E}_{\mathsf{L}_\Phi}^*(\mathcal{H}) \\
&= \mathcal{E}_{\mathsf{L}_\Phi}(\widehat{h}_\lambda) - \widehat{\mathcal{E}}_{\mathsf{L}_\Phi, S}(\widehat{h}_\lambda) + \widehat{\mathcal{E}}_{\mathsf{L}_\Phi, S}(\widehat{h}_\lambda) - \mathcal{E}_{\mathsf{L}_\Phi}^*(\mathcal{H}) \\
&\leq \mathcal{E}_{\mathsf{L}_\Phi}(\widehat{h}_\lambda) - \widehat{\mathcal{E}}_{\mathsf{L}_\Phi, S}(\widehat{h}_\lambda) + \widehat{\mathcal{E}}_{\mathsf{L}_\Phi, S}(h^*) - \mathcal{E}_{\mathsf{L}_\Phi}^*(\mathcal{H}) \\
&\leq \mathcal{E}_{\mathsf{L}_\Phi}(\widehat{h}_\lambda) - \widehat{\mathcal{E}}_{\mathsf{L}_\Phi, S}(\widehat{h}_\lambda) + \widehat{\mathcal{E}}_{\mathsf{L}_\Phi, S}(h^*) - \mathcal{E}_{\mathsf{L}_\Phi}^*(h^*) + \epsilon \\
&\leq 2\left[2\mathfrak{R}_m^\lambda(\mathcal{H}) + B_\lambda\sqrt{\frac{\log(2/\delta)}{2m}}\right] + \epsilon.
\end{aligned}$$

Since the inequality holds for all $\epsilon > 0$, it implies:

$$\mathcal{E}_{\mathsf{L}_\Phi}(\widehat{h}_\lambda) - \mathcal{E}_{\mathsf{L}_\Phi}^*(\mathcal{H}) \leq 4\mathfrak{R}_m^\lambda(\mathcal{H}) + 2B_\lambda\sqrt{\frac{\log(2/\delta)}{2m}}.$$

Substituting this inequality into the $\mathcal{H}$-consistency bound in the assumption completes the proof. $\square$

# G. Proof of Theorem 5.4

Let $B_\Phi$ be an upper bound for the function $\Phi$. This boundedness holds in practice when we consider a bounded input space.

**Lemma G.1.** *Fix $\epsilon > 0$ and assume $|\lambda_1 - \lambda_2| \leq \epsilon$. Then, the following inequalities hold:*

$$\left|\mathfrak{R}_m^{\lambda_1}(\mathcal{H}) - \mathfrak{R}_m^{\lambda_2}(\mathcal{H})\right| \leq \frac{4\max_i|\beta_i|B_\Phi\epsilon}{m}, \quad |B_{\lambda_1} - B_{\lambda_2}| \leq 4\max_i|\beta_i|B_\Phi\epsilon.$$

*Proof.* Let $\mathsf{L}_\Phi^\lambda$ be the surrogate loss (6) with the parameter $\lambda$. By the definition of the cost function (4), the following holds for any $(h, x, y) \in \mathcal{H} \times \mathcal{X} \times \mathcal{Y}$:

$$\left|\mathsf{L}_\Phi^{\lambda_1}(h, x, y) - \mathsf{L}_\Phi^{\lambda_2}(h, x, y)\right| \leq 4\max_i|\beta_i|B_\Phi|\lambda_1 - \lambda_2|.$$

Thus, by definition of Rademacher complexity and using $B_\lambda = \sup_{h,x,y}\mathsf{L}_\Phi(h, x, y)$, we have

$$\left|\mathfrak{R}_m^{\lambda_1}(\mathcal{H}) - \mathfrak{R}_m^{\lambda_2}(\mathcal{H})\right| \leq \frac{4\max_i|\beta_i|B_\Phi\epsilon}{m}, \quad |B_{\lambda_1} - B_{\lambda_2}| \leq 4\max_i|\beta_i|B_\Phi\epsilon.$$

This completes the proof. $\square$

**Theorem 5.4.** *Assume that the surrogate loss $\mathsf{L}_\Phi$ admits a $\overline{\Gamma}$-$\mathcal{H}$-consistency bound with respect to $\ell^\lambda$. Then, for any $\delta > 0$, with probability at least $1 - \delta$ over the draw of a sample $S$ from $\mathcal{D}^m$, the following estimation bound holds for an empirical minimizer $\widehat{h}_S \in \mathcal{H}$ of the surrogate loss $\mathsf{L}_\Phi$ over $S$ and $\lambda \in [\lambda_{\min}, \lambda_{\max}]$:*

$$\mathcal{E}_{\ell^\lambda}(\widehat{h}_S) - \mathcal{E}_{\ell^\lambda}^*(\mathcal{H}) \leq \overline{\Gamma}\left(\mathcal{M}_{\mathsf{L}_\Phi}(\mathcal{H}) + 4\mathfrak{R}_m^\lambda(\mathcal{H}) + \frac{8\max_i|\beta_i|B_\Phi}{m^2} + \left[2B_\lambda + \frac{4\max_i|\beta_i|B_\Phi}{m}\right]\sqrt{\frac{\log\frac{2\Delta\lambda\, m}{\delta}}{2m}}\right) + \frac{\overline{\ell}_\beta}{m}.$$

*where $\Delta\lambda = \lambda_{\max} - \lambda_{\min}$. In particular, when $\overline{\Gamma}(t) = 2(\mathsf{L}_{\max})^{\frac{1}{2}}t^{\frac{1}{2}}$ for $\Phi = \Phi_{\log}$, the bound can be expressed as follows:*

$$\mathcal{E}_{\ell^\lambda}(\widehat{h}_S) - \mathcal{E}_{\ell^\lambda}^*(\mathcal{H}) \leq (2\mathsf{L}_{\max})^{\frac{1}{2}}\left[\mathcal{M}_{\mathsf{L}_\Phi}(\mathcal{H}) + 4\mathfrak{R}_m^\lambda(\mathcal{H}) + \frac{8\max_i|\beta_i|B_\Phi}{m^2} + \left[2B_\lambda + \frac{4\max_i|\beta_i|B_\Phi}{m}\right]\sqrt{\frac{\log\frac{2(\Delta\lambda)\, m}{\delta}}{2m}}\right]^{\frac{1}{2}} + \frac{\overline{\ell}_\beta}{m}.$$

*Proof.* To derive the uniform bound, we cover the interval $[\lambda_{\min}, \lambda_{\max}]$ using sub-intervals of size $1/m$. Consider sequences $(\lambda_k)_k$, where $\lambda_k = \lambda_{\min} + \frac{k-1}{m}$, $1 \le k \le m(\lambda_{\max} - \lambda_{\min})$. By Theorem 5.3 and the standard uniform bounds (Mohri et al., 2018), for any $\delta > 0$, with probability at least $1 - \delta$, the following bound holds for all $k$:

$$\mathcal{E}_{\lambda_k}(\widehat{h}_S) - \mathcal{E}_{\lambda_k}^*(\mathcal{H}) \le \Gamma\left(\mathcal{M}_{\mathsf{L}_\Phi}(\mathcal{H}) + 4\mathfrak{R}_m^{\lambda_k}(\mathcal{H}) + 2B_{\lambda_k}\sqrt{\frac{\log \frac{2(\lambda_{\max} - \lambda_{\min})m}{\delta}}{2m}}\right).$$

Then, for any $\lambda \in [\lambda_{\min}, \lambda_{\max}]$, there exists $k \ge 1$ such that $|\lambda - \lambda_k| \le \frac{1}{2m}$. By Lemma E.1 and Lemma G.1, we obtain

$$\mathcal{E}_{\ell^\lambda}(\widehat{h}_S) - \mathcal{E}_{\ell^\lambda}^*(\mathcal{H}) \le \overline{\Gamma}\left(\mathcal{M}_{\mathsf{L}_\Phi}(\mathcal{H}) + 4\mathfrak{R}_m^\lambda(\mathcal{H}) + \frac{8\max_i|\beta_i|B_\Phi}{m^2} + \left(2B_\lambda + \frac{4\max_i|\beta_i|B_\Phi}{m}\right)\sqrt{\frac{\log \frac{2(\lambda_{\max} - \lambda_{\min})m}{\delta}}{2m}}\right) + \frac{\overline{\ell}_\beta}{m}.$$

This completes the proof of the first bound. The second bound follows directly by substituting $\overline{\Gamma}(t) = 2(\mathsf{L}_{\max})^{\frac{1}{2}}t^{\frac{1}{2}}$ into the first bound. $\square$

## H. Proof of Theorem 5.5

**Theorem 5.5.** *Let $\epsilon = \frac{\epsilon_m}{2\underline{\ell}_\beta}$. For any $\delta > 0$, with probability at least $1 - \delta$, Algorithm 2 returns in $O\left(\log_2\left(\frac{\lambda_{\max} - \lambda_{\min}}{\epsilon}\right)\right)$ time a hypothesis $\widehat{h}_\lambda$ that admits the following guarantee:*

$$\mathcal{L}(\widehat{h}_\lambda) \le \mathcal{L}^*(\mathcal{H}) + \frac{4\epsilon_m}{\underline{\ell}_\beta}.$$

*Proof.* Suppose that $\widehat{h}_\lambda$ is the solution returned at step 10 of Algorithm 2. Then, by definition of the algorithm, we must have $|\widehat{\mathcal{E}}_{\ell^\lambda}(\widehat{h}_\lambda)| \le \epsilon_m$, which, by the standard generalization bound (Mohri et al., 2018), implies that (with high probability)

$$\mathcal{E}_{\ell^\lambda}(\widehat{h}_\lambda) \le 2\epsilon_m$$

Expanding the definition of the surrogate risk, this gives

$$\mathbb{E}[\ell_\alpha(h, x, y)] - \lambda \mathbb{E}[\ell_\beta(h, x, y)] \le 2\epsilon_m.$$

Dividing through by $\mathbb{E}[\ell_\beta(h, x, y)]$ yields

$$\mathcal{L}(\widehat{h}_\lambda) \le \lambda + \frac{2\epsilon_m}{\mathbb{E}[\ell_\beta(h, x, y)]} \le \lambda + \frac{2\epsilon_m}{\underline{\ell}_\beta}. \tag{7}$$

Next, observe that for all $h \in \mathcal{H}$, we have

$$\mathcal{E}_{\ell^\lambda}(h) - \mathcal{E}_{\ell^{\lambda*}}(h) = (\lambda - \lambda^*)\mathbb{E}[\ell_\beta(h, x, y)].$$

Rearranging gives

$$(\lambda - \lambda^*) = \frac{\mathcal{E}_{\ell^\lambda}(h) - \mathcal{E}_{\ell^{\lambda*}}(h)}{\mathbb{E}[\ell_\beta(h, x, y)]}.$$

Since $\mathcal{E}_{\ell^{\lambda*}}(\mathcal{H}) = 0$, by definition of the infimum, it follows that $\mathcal{E}_{\ell^{\lambda*}}(h) \ge 0$. In view of that, we can write:

$$(\lambda - \lambda^*) \le \frac{\mathcal{E}_{\ell^\lambda}(h)}{\mathbb{E}[\ell_\beta(h, x, y)]}.$$

Applying this inequality with $h = \widehat{h}_\lambda$ and substituting into (7), we obtain

$$\mathcal{L}(\widehat{h}_\lambda) \le \lambda^* + \frac{2\epsilon_m}{\underline{\ell}_\beta} + \frac{\mathcal{E}_{\ell^\lambda}(\widehat{h}_\lambda)}{\mathbb{E}[\ell_\beta(\widehat{h}_\lambda, x, y)]} \le \lambda^* + \frac{4\epsilon_m}{\underline{\ell}_\beta}.$$

This ends the analysis of that case.

Now, consider the case where the algorithm terminates with $|b - a| \leq \epsilon_m$. In this case, we have (with high probability) $|\lambda - \lambda^*| \leq \epsilon$. By Lemma E.1, this implies

$$\mathcal{E}^*_{\ell^\lambda}(\mathcal{H}) \leq \epsilon \overline{\ell}_\beta.$$

Combining this with the estimation bound for $\ell^\lambda$, we have (with high probability)

$$\mathcal{E}_{\ell^\lambda}(\widehat{h}_\lambda) \leq \mathcal{E}^*_{\ell^\lambda}(\mathcal{H}) + \epsilon_m \leq \epsilon \overline{\ell}_\beta + \epsilon_m.$$

Thus,

$$\mathbb{E}[\ell^\lambda_{\boldsymbol{\alpha}}(\widehat{h}_\lambda, x, y)] - \lambda \, \mathbb{E}[\ell^\lambda_\beta(\widehat{h}_\lambda, x, y)] \leq \epsilon \overline{\ell}_\beta + \epsilon_m.$$

Dividing by $\mathbb{E}[\ell^\lambda_\beta(\widehat{h}_\lambda, x, y)]$ both sides yields

$$\mathcal{L}_{\alpha,\beta}(\widehat{h}_\lambda) \leq \lambda + \frac{\epsilon \overline{\ell}_\beta + \epsilon_m}{\underline{\ell}_\beta} \leq \lambda^* + \epsilon + \frac{\epsilon \overline{\ell}_\beta + \epsilon_m}{\underline{\ell}_\beta}.$$

Choosing $\epsilon = \epsilon_m / (2\overline{\ell}_\beta)$ yields

$$\mathcal{L}_{\alpha,\beta}(\widehat{h}_\lambda) \leq \lambda + \frac{\epsilon \overline{\ell}_\beta + \epsilon_m}{\underline{\ell}_\beta} \leq \lambda^* + \frac{\epsilon_m}{2\overline{\ell}_\beta} + \frac{3\epsilon_m}{2\underline{\ell}_\beta} \leq \lambda^* + \frac{2\epsilon_m}{\underline{\ell}_\beta}.$$

This completes the proof. $\qquad\square$

## I. Proof of Theorem 5.6

**Theorem 5.6.** *For any $\delta > 0$, with probability at least $1 - \delta$, for $\epsilon \leq \frac{\epsilon_m}{2\overline{\ell}_\beta}$, Algorithm 3 returns in $O\left(\frac{\lambda_{\max} - \lambda_{\min}}{\epsilon}\right)$ time a hypothesis $\widehat{h}_\lambda$ that admits the following guarantee:*

$$\mathcal{L}(\widehat{h}_\lambda) \leq \mathcal{L}^*(\mathcal{H}) + \frac{2\epsilon_m}{\underline{\ell}_\beta}.$$

*Proof.* Since the algorithm terminates with $a + i\epsilon > b$, we have $|\lambda - \lambda^*| \leq \epsilon$. By Lemma E.1, this implies

$$\mathcal{E}^*_{\ell^\lambda}(\mathcal{H}) \leq \epsilon \overline{\ell}_\beta.$$

Combining this with the estimation bound for $\ell^\lambda$, we have (with high probability)

$$\mathcal{E}_{\ell^\lambda}(\widehat{h}_\lambda) \leq \mathcal{E}^*_{\ell^\lambda}(\mathcal{H}) + \epsilon_m \leq \epsilon \overline{\ell}_\beta + \epsilon_m.$$

Thus,

$$\mathbb{E}[\ell^\lambda_{\boldsymbol{\alpha}}(\widehat{h}_\lambda, x, y)] - \lambda \, \mathbb{E}[\ell^\lambda_\beta(\widehat{h}_\lambda, x, y)] \leq \epsilon \overline{\ell}_\beta + \epsilon_m.$$

Dividing by $\mathbb{E}[\ell^\lambda_\beta(\widehat{h}_\lambda, x, y)]$ both sides yields

$$\mathcal{L}_{\alpha,\beta}(\widehat{h}_\lambda) \leq \lambda + \frac{\epsilon \overline{\ell}_\beta + \epsilon_m}{\underline{\ell}_\beta} \leq \lambda^* + \epsilon + \frac{\epsilon \overline{\ell}_\beta + \epsilon_m}{\underline{\ell}_\beta}.$$

Choosing $\epsilon \leq \epsilon_m / (2\overline{\ell}_\beta)$ yields

$$\mathcal{L}_{\alpha,\beta}(\widehat{h}_\lambda) \leq \lambda + \frac{\epsilon \overline{\ell}_\beta + \epsilon_m}{\underline{\ell}_\beta} \leq \lambda^* + \frac{\epsilon_m}{2\overline{\ell}_\beta} + \frac{3\epsilon_m}{2\underline{\ell}_\beta} \leq \lambda^* + \frac{2\epsilon_m}{\underline{\ell}_\beta}.$$

This completes the proof. $\qquad\square$

