# OpenReview forum: "Principled Algorithms for Optimizing Generalized Metrics in Binary Classification"
_ICML.cc/2025/Conference — ICML 2025 poster_

### Official Review · Reviewer_dynv · 2025-03-04

**Overall Recommendation:** 5

**Summary:**

This paper studies the problem of optimizing a broad class of metrics used in class imbalance or class asymmetric scenarios. Previous approaches rely on threshold-based methods that approximates Bayes-optimal classifiers with guarantees of consistency which is asymptotic. This paper first shows that optimizing such kind of metrics can be interpreted into the problem of optimizing an alternative cost-sensitive loss, provided an oracle access of the loss value of the best hypothesis in hypothesis set. The authors then shows that the cost-sensitive loss of the above form can all be optimized with non-asymptotic theoretical guarantee via the technique of surrogate loss. Meanwhile, the oracle access required to compute the loss can also be relaxed by leveraging binary search using Rademacher bounds as separating condition. Experimental results show the effectiveness of the proposed approach.

**Claims And Evidence:**

Yes.

**Essential References Not Discussed:**

No.

**Experimental Designs Or Analyses:**

Yes.

**Methods And Evaluation Criteria:**

Yes.

**Other Comments Or Suggestions:**

No.

**Other Strengths And Weaknesses:**

- Strength: The problem investigated in the paper is fundamental. The solution is intuitive and make sense, and specifically, using Rademacher bounds to make binary search for metric optimization is a nice idea. The structure of the paper is clear, each section covers a relatively self-contained step.
- Weakness: I do not have concern on weakness.

**Questions For Authors:**

(1) The authors propose to leverage Rademacher bounds, which can be approximated via classical learning process such as ERM. I am curious on the inherent difficulty in the metric optimization step (i.e., finding $\lambda^*$) in this paper. For example, is metric optimization at least as difficult as supervised learning multiple times with alternative losses that are independent of the metric learning process? Anyway, the answer to this question may not let me increase my evaluation to this paper, since it cannot be increased anymore.

**Relation To Broader Scientific Literature:**

The authors propose a novel approach to deal with class-imbalanced and cost-sensitive binary classification problems, which is fundamental in many other literatures.

**Theoretical Claims:**

Yes.

---

> ### Author Rebuttal · Authors · 2025-03-30
>
> We thank the reviewer for their strong support of our work. Below please find responses to specific questions.
>
> **1. Questions: The authors propose to leverage Rademacher bounds, which can be approximated via classical learning process such as ERM. I am curious on the inherent difficulty in the metric optimization step (i.e., finding $\lambda^\*$) in this paper. For example, is metric optimization at least as difficult as supervised learning multiple times with alternative losses that are independent of the metric learning process? Anyway, the answer to this question may not let me increase my evaluation to this paper, since it cannot be increased anymore.**
>
> **Response:** This is a natural question. The reviewer is asking about the difficulty of directly optimizing a general metric defined as the ratio of the expectations of two loss functions, both linear in ${\mathsf h}$, where ${\mathsf h}(x) = \text{sign}(h(x))$.
>
> While the metric is quasi-concave in ${\mathsf h}$, its optimization with respect to $h$ is NP-hard (even with a constant denominator), even for a linear hypothesis set. In contrast, each surrogate loss optimization problem we consider (framed as supervised learning) can be solved in polynomial time over a convex hypothesis set, as the surrogate loss functions we adopt are convex.
>
> Moreover, directly optimizing the empirical ratio of the numerator and denominator may not yield a provably good approximation of the metric, since their expectation does not align with the ratio of expectations.
>
> We will include this discussion in the final version. If our interpretation of the reviewer’s question is incorrect, please clarify, and we will be happy to address it.

---

### Official Review · Reviewer_wfsZ · 2025-03-10

**Overall Recommendation:** 3

**Summary:**

This paper proposes METRO for generalized metric optimization in binary classification. The authors reformulate metric optimization as a generalized cost-sensitive learning problem, and introduce a new family of surrogate loss functions. They theoretically prove the $\mathcal{H}$-consistency guarantees for theses losses and develop finite-sample learning bounds for the proposed algorithm. Experiments  on image classification demonstrate that their algorithm outperforms other baselines.

**Claims And Evidence:**

Yes

**Essential References Not Discussed:**

I am not very familiar with Bayes learning and related works are well discussed as far as I am concerned.

**Experimental Designs Or Analyses:**

Yes

**Methods And Evaluation Criteria:**

Yes

**Other Comments Or Suggestions:**

See Questions.

**Other Strengths And Weaknesses:**

Strengths
1. The motivation and the introduction to the generalized metrics optimization framework are clear.
2. The reformulation of generalized cost-sensitive learning and the derivation of finite-sample learning bounds seem novel to me.

Weaknesses
1. The experiments were limited to the classic image classification task. Algorithms were not applied to scenarios such as significant class imbalance, which is a fundamental motivation for this paper as stated in Section 1.

**Questions For Authors:**

I am curious about how the runtime of the proposed method compares to other baselines.

**Relation To Broader Scientific Literature:**

The proposed framework can be used to design algorithms with finite-sample learning bounds, unlike Bayes-consistency.

**Theoretical Claims:**

Proofs seem correct to me.

---

> ### Author Rebuttal · Authors · 2025-03-30
>
> Thank you for your encouraging review. We will take your suggestions into account when preparing the final version. Below please find responses to specific questions.
>
> **1. Weaknesses: The experiments were limited to the classic image classification task. Algorithms were not applied to scenarios such as significant class imbalance, which is a fundamental motivation for this paper as stated in Section 1.**
>
> **Response:** Thank you for the valuable suggestion. Following previous work, we used standard image classification datasets for our empirical evaluation to demonstrate the effectiveness of our methods compared to prior baselines. However, we agree that evaluating our algorithms on more imbalanced datasets would be beneficial, and we plan to include such experiments and comparisons in the final version.
>
> **2. Questions: I am curious about how the runtime of the proposed method compares to other baselines.**
>
> **Response:** The per-epoch computational cost of our method is comparable to that of Algorithm 2 in Koyejo et al. (2014). Both methods involve a single hyperparameter, and for a fixed value of this parameter, the computational cost is similar to training a standard binary classifier using a standard surrogate loss. We will include a more in-depth discussion of this topic in the relevant section of the paper.

---

### Official Review · Reviewer_TFXy · 2025-03-12

**Overall Recommendation:** 4

**Summary:**

This article introduces a novel optimization approach for generalized metrics in binary classification. The primary method involves converting the fractional form into a summation form. However, the method introduces a parameter, $\lambda$, whose optimal value is unknown and requires estimation, indicating that cross-validation may be necessary to identify the appropriate parameter.

### update after rebuttal

The authors have provided explanations for the theoretical technical details I raised and discussed the limitations of their approach. They have also clarified aspects of their experimental design and incorporated relevant discussions of the literature I suggested.

Overall, I appreciate the theoretical results and motivation behind the authors' proposed method. However, from practical perspective, the approach does have certain limitations, such as the selection of $\lambda$ and/or the sacrifice of data for cross-validation. Additionally, more comparisons with existing consistency methods in the literature will be better. Nevertheless, I believe this paper is worthy of publication. Therefore, I maintain my current rating.

**Claims And Evidence:**

Yes, the method has been validated through both theoretical analysis and experimental results. I have included some comments in the following comment boxes.

**Essential References Not Discussed:**

The paper systematically discusses the literature, and I would like to add few references.

### Bayes-rule for F-score

Jansche, M. (2007, June). A maximum expected utility framework for binary sequence labeling. In Proceedings of the 45th Annual Meeting of the Association of Computational Linguistics (pp. 736-743).

Dai, B., & Li, C. (2023). RankSEG: a consistent ranking-based framework for segmentation. Journal of Machine Learning Research, 24(224), 1-50.

### Threshold study for F-score

Lipton, Z. C., Elkan, C., & Narayanaswamy, B. (2014). Thresholding classifiers to maximize F1 score. arXiv preprint arXiv:1402.1892.

**Experimental Designs Or Analyses:**

The experimental design is reasonable, but it could be further improved or clarified. Please refer to the detailed comments in the previous comment box.

**Methods And Evaluation Criteria:**

### **Method**

Overall, the methods presented in the article are motivated by several theorems, primarily Theorem 3.1 and Theorem 5.1, which I find to be a particularly interesting aspect of the work.

I would like to double-check the estimation of $\lambda$ using the proposed method.

In both Algorithms 2 and 3, $\hat{\mathcal{E}_{l^{\lambda}}}$ and $\hat{h}$ are estimated based on the same dataset, will this approach be effective?

Moreover, is it valid to derive the result of Theorems 5.4 and 5.5? I can understand if the data were different.

Specifically, in the proof of Theorem 5.4,

$$
\hat{\mathcal{E}_{l^{\lambda}}}(\hat{h}) \leq \epsilon,
$$

implies

$$
\mathcal{E}_{l^{\lambda}}(\hat{h}) \leq 2\epsilon.
$$

Consider the case of an overparameterized model $h$ such that $\hat{\mathcal{E}}_{l^{\lambda}}(\hat{h})$ is pretty small;

but, in this scenario, it appears that $\mathcal{E}_{l^{\lambda}}(\hat{h})$ could still be large.

### **Experiments**

The experimental section could be improved.

Some statements lack clarity; for instance, it is unclear whether the METRO Algorithm ultimately employs Algorithm 2 or Algorithm 3. Additionally, since the datasets are generated from multiclass datasets into binary datasets, providing more details would be beneficial.

The model only utilizes a three-hidden-layer CNN with ReLU activations; incorporating additional models could enhance the overall convincingness of the results.

**Other Comments Or Suggestions:**

N.A.

**Other Strengths And Weaknesses:**

N.A.

**Questions For Authors:**

N.A.

**Relation To Broader Scientific Literature:**

I do not fully understand why the proposed method outperforms Koyejo et al. (2014). Both approaches theoretically present correct methods with an additional parameter; it seems to me that the difference lies primarily in how this parameter is tuned. I currently cannot grasp why the proposed algorithm has an advantage in tuning this parameter.

**Theoretical Claims:**

I reviewed the proofs of Theorems 3.1 and 5.1, as well as part of Theorem 5.4.

Theorems 3.1 and 5.1 are correct;

I have posed some questions regarding Theorem 5.4 in the previous comment box.

---

> ### Author Rebuttal · Authors · 2025-03-30
>
> Thank you for your appreciation of our work. We will take your suggestions into account when preparing the final version. Below please find responses to specific questions.
>
> **1. Method:** Thank you for the insightful comments. The quantities $\hat{\mathcal{E}} _{\ell^{\lambda}}$ and $\hat{h}$ are approximated using data sampled from the same distribution, but they can be obtained from different samples. In practice, as done in Koyejo et al. (2014), we can split the training data into two parts: $\hat{\lambda}$ is obtained from one part, and then used to train the hypothesis $\hat{h} _{\hat{\lambda}}$ on the other.
>
> Theorems 5.4 and 5.5 remain valid as long as the data are sampled independently from the same distribution. Regarding your observation: in the case of an overparameterized model, it is true that $\mathcal{E} _{\ell^{\lambda}}(\hat{h} _{\lambda}) \leq 2\epsilon_m$ still holds by the standard generalization bound (Mohri et al., 2018), but the value of $\epsilon_m$ may be larger due to the high complexity of the model. As you correctly noted, $\epsilon_m$ becomes small only when the sample size is sufficiently large relative to the complexity of the hypothesis set. This limitation applies broadly to most generalization bounds for complex neural networks.
>
> The current analysis of overparameterized settings typically requires alternative tools, particularly those that account for the optimization algorithm (e.g., SGD) and its dynamics. Such analyses often apply only to more restricted model families.
>
> We will elaborate on these points and revise the presentation of Algorithms 2 and 3 for improved clarity in the final version.
>
> **2. Experiments:** Thank you for the valuable suggestions. As indicated on line 435 (left), we used the METRO Algorithm 3 in our experiments. We will follow the reviewer’s recommendation to add more details about the datasets and include additional experimental results using alternative models in the final version. We also plan to incorporate experiments on more imbalanced datasets, as suggested by Reviewer wfsZ.
>
> **3. Relation To Broader Scientific Literature:** First, we should emphasize that the guarantees for these methods differ fundamentally. Our METRO algorithms for optimizing general metrics are backed by strong theoretical guarantees that apply to arbitrary hypothesis sets and include finite-sample bounds. In contrast, prior methods (Koyejo et al., 2014) provide only Bayes-consistency guarantees, which hold solely for the class of all measurable functions and offer no convergence bound. Moreover, their approach lacks convergence rate guarantees for parameter tuning, unlike our finite-sample bounds.
>
> Beyond these theoretical advantages, a key limitation of prior methods is their dependence on the structure of the Bayes-optimal solution. Since the Bayes-optimal predictor for a given metric often differs from binary classification only by an offset, their approach first trains a binary classifier and then selects an optimal threshold or offset. However, this strategy fails when the best predictor within a restricted hypothesis set does not align with the Bayes-optimal form (see Figure 1). Consequently, irrespective of parameter estimation or tuning, their approach does not succeed in general, as our example illustrates.
>
> In contrast, our approach and algorithms provide convergence rate guarantees for arbitrary hypothesis sets and do not rely on the specific form of the Bayes-optimal solution, ensuring robust and theoretically grounded optimization.
>
> **4. Essential References Not Discussed:** We thank the reviewer for pointing out these relevant references on optimizing the F-score. They are indeed closely related to our work, and we will be sure to include and discuss them in the final version.

---

### Decision · Program_Chairs · 2025-05-01

**Decision:**

Accept (poster)

**Comment:**

This paper introduces a novel method for optimizing generalized metrics in binary classifications. The theoretical foundation is backed by H-consistency and uniform convergence results. Additionally, the proposed method demonstrates strong empirical performance across various datasets.

Overall, the reviewers have provided consistently positive feedback regarding this paper, and I recommend its acceptance. Nevertheless, the authors are strongly encouraged to include all necessary details in their rebuttal in the final version.